# HIF sustain a transcriptional regulatory circuit of *EPAS1* expression in renal clear cell carcinoma

Stephanie Naas [1], René Krüger[1], Steffen Grampp[1], Victoria Lauer [1], Andre Kraus[1], Julia Naas [2], Fabian Müller [3], Franziska Gsottberger [3], Mario Schiffer [1], Bernd Wullich[4,5], Arndt Hartmann[5,6], Marc P. Stemmler [7] & Johannes Schödel [1,5] ✉

Initiation and sustainment of oncogenic signaling is a hallmark of cancer evolution and progression. In renal clear cell carcinoma, loss of von Hippel-Lindau protein causes stabilization of hypoxia-inducible transcription factors (HIF) evoking a pseudo-hypoxic response, perturbing epithelial homeostasis and leading to cancer development. Although genetic polymorphisms link the *EPAS1* oncogene (coding for HIF-2α) to renal cancer and anti-HIF-2 compounds emerge as renal tumor therapies, little is known about transcriptional dysregulation of this factor in renal malignancies. We use genetic, epigenetic and transcriptomic data from large patient cohorts and cell models to dissect mechanisms of augmented *EPAS1* transcription in clear cell renal cell carcinoma. We define an oncogenic enhancer of *EPAS1* which operates depending on the presence of HIF and renal lineage-specific factors, thereby providing evidence for an auto-regulatory feed-forward circuit of HIF-2α regulation which promotes renal cancer growth.

Biallelic mutations in the von Hippel-Lindau (*VHL*) tumor suppressor gene are truncal events leading to the development of clear cell renal cell carcinoma (ccRCC)[1–4]. ccRCC is the most common type of renal cancer, accounting for more than 70% of kidney tumors[5]. *VHL* encodes pVHL, a member of the E3 ubiquitin ligase complex, which functions as an adapter molecule for the complex to recognize hydroxylated α-subunits of hypoxia inducible transcription factors (HIF)[6,7]. Loss of pVHL induces stabilization of HIF-α protein independent of the presence of molecular oxygen and triggers its broad transcriptional activity.

In most tumors, activation of HIF-2 signaling either by hypoxia or genetic mutations promotes tumor growth and correlates with adverse outcomes[8]. With respect to ccRCC, HIF-2α overexpression led to accelerated cancer growth in xenograft models and the transcriptional output of HIF-2 - as opposed to targets of HIF-1 - was associated with a negative outcome in the KIRC (Kidney renal clear cell carcinoma) cohort in The Cancer Genome Atlas Program (TCGA)[9–11]. Furthermore, single nucleotide polymorphisms (SNP) in intronic regions of the Endothelial PAS domain-containing protein 1 (*EPAS1*) gene, which codes for HIF-2α, predispose to the development of renal

[1]Department of Nephrology and Hypertension, Uniklinikum Erlangen and Friedrich-Alexander- Universität Erlangen-Nürnberg, Erlangen, Germany. [2]Center for Integrative Bioinformatics Vienna (CIBIV), Max Perutz Labs, University of Vienna and Medical University of Vienna, Vienna BioCenter, Vienna, Austria. [3]Department of Internal Medicine 5—Hematology and Oncology, Uniklinikum Erlangen and Friedrich Alexander University Erlangen-Nuremberg (FAU), Erlangen, Germany. [4]Department of Urology and Pediatric Urology, Uniklinikum Erlangen and Friedrich-Alexander- Universität Erlangen-Nürnberg, Erlangen, Germany. [5]Comprehensive Cancer Center Erlangen-EMN (CCC ER-EMN), Erlangen, Germany. [6]Institute of Pathology, Uniklinikum Erlangen and Friedrich-Alexander-Universität Erlangen-Nürnberg, Erlangen, Germany. [7]Department of Experimental Medicine 1, Nikolaus-Fiebiger Center for Molecular Medicine, Friedrich-Alexander University of Erlangen-Nürnberg, Erlangen, Germany. ✉e-mail: johannes.schoedel@uk-erlangen.de

cancer[12]. Thus, *EPAS1* is regarded as an oncogene in ccRCC and has emerged as a promising therapeutic target[13]. HIF-2 inhibitors have been licensed in the United States and Europe for the treatment of VHL-associated cancers not requiring immediate surgery in patients with hereditary VHL-disease and of patients with progressed stages of sporadic RCC[14–16]. Interestingly, preclinical data suggested that higher levels of HIF-2α mRNA and protein within tumors render them more sensitive to anti-HIF-2 therapy[17,18].

HIF-2α protein could not be detected in the non-diseased renal epithelium and mechanisms how HIF-2α expression was released in early tubular lesions upon VHL-loss have not been defined yet, but may involve transcriptional, posttranscriptional and posttranslational mechanisms[19,20]. The presence of non-coding RCC-associated SNPs at the *EPAS1* locus indicates that transcriptional dysregulation may contribute significantly to this process. Genome-wide association studies (GWAS) and fine-mapping analyses have defined two independent genetic signals within intronic regions of the *EPAS1* gene which are associated with the development of renal cancer[12,21,22]. Of note, for one signal comprising the variant rs12617313 a significant association with the occurrence of *VHL* alterations in the corresponding renal tumors was determined suggesting an interaction of common genetic variation with tumor mutations[21]. Although the SNPs within the *EPAS1* gene generated one of the strongest genome-wide association signals in RCC, surprisingly little is known about the regulation of HIF-2α mRNA expression in general and specifically in the context of renal cancer[12]. In this respect, DNA demethylation via inhibition of DNA (cytosine-5)-methyltransferase 3 A (*DNMT3A*) of the *EPAS1* promoter has been shown to be involved in releasing HIF-2α expression in renal tumors[23]. However, genetic mutations of *DNMT3A* are rare in kidney cancer indicating that other mechanisms participate in forcing HIF-2α release.

Recently, renal lineage-specific transcription factors such as paired-box-protein 8 (PAX8) and hepatocyte nuclear factor 1-beta (HNF-1β) have been identified as significant drivers of oncogenic signaling in kidney cancer explaining some of the tissue specificity of the tumors[24,25]. For example, PAX8 occupied enhancers in ccRCC cells which control the expression of metabolic genes[24]. Moreover, pro-proliferative PAX8 signaling co-operated with HIF signaling at a variety of regulatory DNA elements including the renal cancer susceptibility locus at a *CCND1* enhancer[25]. PAX8 and HNF-1β form core regulatory circuitries in renal cancer[24]. These may stem from regulatory loops preserved from the renal tubule progenitor cells or may be rewired within renal cancer cells incorporating other oncogenic signals such as HIF.

The presence of HIF-2α in early tumorous lesions in the kidney and its susceptibility to therapy together with the fact that in most cases renal cancer evolution takes decades opens a broad window of opportunity for interventions[19,26]. Taking advantage of this window requires a much better understanding of the transcriptional regulation of HIF-2α with regard to different genetic backgrounds as well as its transcriptional consequences and interactions.

In this work, we define transcriptional enhancers of *EPAS1* expression in ccRCC. We explore their effects on HIF-2α mRNA levels in the context of genetic susceptibility to RCC, their interactions with lineage-specific transcription factors, and their downstream effects on ccRCC biology.

## Results

### ccRCC tumor cells overexpress HIF-2α mRNA
Stabilization of HIF-2α protein is a hallmark of VHL-defective ccRCC. In addition, genetic variation in non-protein coding regions at the *EPAS1* locus is associated with the development of ccRCC indicating the involvement of mechanisms upstream of protein-regulation in conferring the cancer risk[12]. In order to screen for a HIF-2α mRNA dysregulation, we analyzed samples from the Erlangen RCC cohort, which comprises 141 tumors and corresponding normal kidney samples from renal cancer patients ($n = 114$ clear cell, $n = 16$ papillary, $n = 11$ chromophobe). By qPCR, we measured significantly increased HIF-2α mRNA levels in ccRCC tissue when compared to corresponding normal tissue (Fig. 1a). In contrast, HIF-2α mRNA was reduced in papillary and in chromophobe RCCs. Similar results were observed in the published TCGA RNA-seq data sets for KIRC (Kidney renal clear cell carcinoma, $n = 72$), KIRP (Kidney renal papillary cell carcinoma, $n = 31$) and KICH (Kidney chromophobe, n = 23) (Fig. 1b). Across all TCGA tumor entities, expression levels of HIF-2α mRNA were highest in the KIRC samples (Supplementary Fig. 1)[27]. These findings suggest that ccRCC-specific mechanisms operate on HIF-2α mRNA expression.

Due to the excessive production of angiogenic factors caused by HIF-stabilization, ccRCC tumors are highly vascularized. HIF-2α is abundantly expressed in endothelial cells[28]. Therefore, we wanted to explore whether the effect of increased HIF-2α mRNA levels in ccRCC tissue was generated from a relatively large number of endothelial cells present in the tumor specimens or directly from tumor cells. First, we performed RNAscope experiments for HIF-2α mRNA in ccRCC tissue from three patients and confirmed substantial levels of HIF-2α transcripts in both, endothelial and cancer cells within the tumor specimens (Fig. 1c, Supplementary Figs. 2 and 3). Second, in a recent single nucleus RNA-seq analysis of ccRCC and normal tissue[29], tumor cells from many individuals exhibited higher levels of HIF-2α and HIF target gene transcripts (e.g., *CCND1, NDRG1, VEGFA*) compared to proximal tubule cells, which are considered as cells of origin for ccRCC (Fig. 1d). Finally, we resorted to freshly explanted ccRCC and corresponding normal tubule cells from 34 patients (Fig. 1e). In the tumor cells, we observed robust HIF protein and absent pVHL in immunoblot experiments (Fig. 1f, Supplementary Fig. 4a). By qPCR, we detected significantly increased levels of HIF-2α mRNA and HIF target gene mRNA (*CCND1, VEGFA*) in tumor cells when compared to corresponding tubule cells from the same individual (Fig. 1g, Supplementary Fig. 4b, c). This indicates that tumor cells contribute substantially to the higher HIF-2α mRNA levels in ccRCC. It also implies that HIF-2α mRNA is subjected to upregulation in ccRCC and that the mechanisms engaged in this regulation are different compared to other tumors, especially other renal tumor entities.

### Epigenetics defines a ccRCC-activated *EPAS1*-enhancer
To gain insight into the mechanisms causing increased HIF-2α mRNA levels in renal tumor cells, we explored the *EPAS1* locus for epigenetic differences between ccRCC and normal tissue or other (renal) tumor entities, respectively. First, we examined isolated primary VHL-defective ccRCC tumor cells and corresponding tubule cells from three individuals (Fig. 1f, Supplementary Fig. 4a). We generated ATAC-seq and H3K27ac ChIP-seq as well as RNA-seq data from these cells. Differential expression analysis of the RNA-seq data confirmed that HIF-2α mRNA was increased in tumor compared to tubule cells (Supplementary Fig. 5a, Supplementary Data 1). Gene set enrichment analysis using a Meta-HIF ChIP-seq data set integrating previously defined HIF-binding sites in ccRCC confirmed that the HIF-pathway is activated in the tumor cells (Supplementary Fig. 5b, Supplementary Data 2). Moreover, analysis of chromatin accessibility followed by motif discovery revealed enrichment of the HIF-binding motif within differentially accessible and active regions in the tumor cells indicating that genomic regions relevant for the HIF-response are preferentially active in tumor cells (Supplementary Fig. 5c, Supplementary Data 3). We expanded our analyses by using the activity-by-contact-model (ABC) of enhancer-promoter regulation in order to identify shared or differentially active enhancer-promoter interactions between tubule and tumor cells on a genome-wide level[30]. This analysis integrated the multi-omics data mentioned above and revealed approximately 65,000 predicted enhancer-promoter interactions per primary tubule cell culture and approximately 63,000 predicted contacts in the corresponding tumor cells (Supplementary Data 4). Between 65–70%

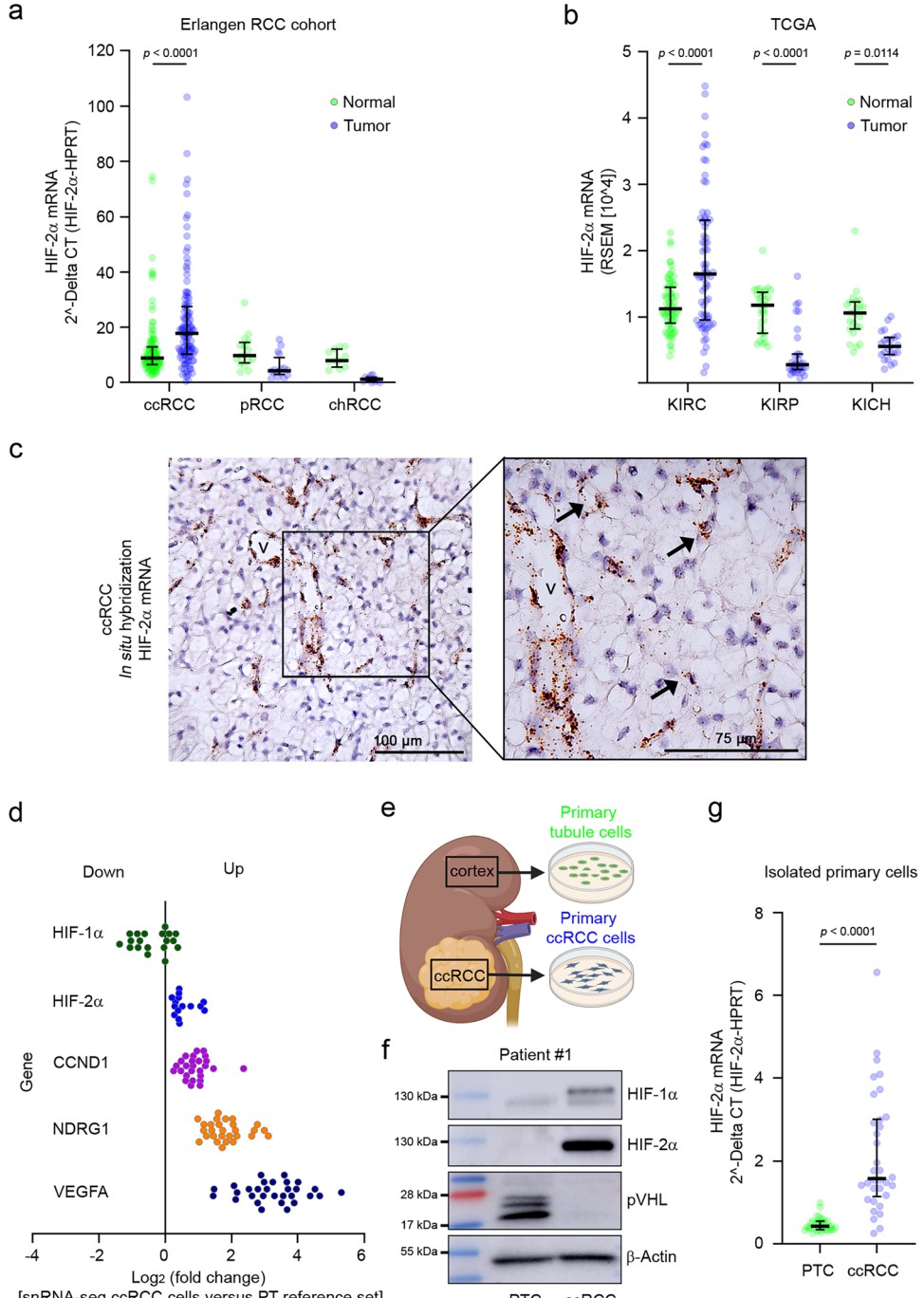

**Fig. 1 | HIF-2α mRNA is overexpressed in ccRCC cells. a** Expression qPCR analysis for HIF-2α mRNA in tissue lysates from renal tumors or corresponding normal kidney tissue from the Erlangen RCC cohort (ccRCC: clear cell renal carcinoma [$n = 114$], pRCC: papillary renal cell carcinoma [$n = 16$], chRCC: chromophobe renal cell carcinoma [$n = 11$]). Data shown are median and interquartile range. Significance was tested by two-way ANOVA followed by Bonferroni's post hoc test. **b** RNA-seq expression values for HIF-2α in different renal tumors and corresponding normal kidney tissue from the TCGA KIRC ($n = 72$), KIRP ($n = 31$) and KICH ($n = 23$) data sets. Median and interquartile range. *P*-values were determined by two-way ANOVA followed by Bonferroni's post hoc test. **c** RNAscope experiment for HIF-2α mRNA in ccRCC tissue. Arrows indicate HIF-2α mRNA transcripts in tumor cells. V = vessel. Scale bars, 100 μm or 75 μm as indicated. Representative image from independent samples of seven different patients with similar results.

**d** Log$_2$ fold change of HIF-related genes in snRNA-seq data comparing tumor cells from 30 ccRCC samples versus proximal tubule (PT) cells from four adjacent kidney tissue samples[29]. Only samples with significantly differentially expressed transcripts are shown (adjusted $p < 0.05$, average log$_2$ fold change ≠ 0). **e** Isolation of renal tumor cells and corresponding tubule cells from non-diseased tissue of tumor nephrectomy specimens. Created in BioRender. Naas, S. (2025) https://BioRender.com/c87s173. **f** Immunoblot analyses for HIF-1α, HIF-2α, pVHL, and β-actin in lysates from primary tubule (PTC) and tumor (ccRCC) cells isolated from tissue specimens of patient #1. Representative blot from 19 independent experiments with similar results. **g** Expression qPCR analysis for HIF-2α mRNA in lysates from primary tubule cells (PTC) and primary tumor (ccRCC) isolated from 34 different individuals. Graph shows median with interquartile range and *p*-values from a two-tailed unpaired t-test.

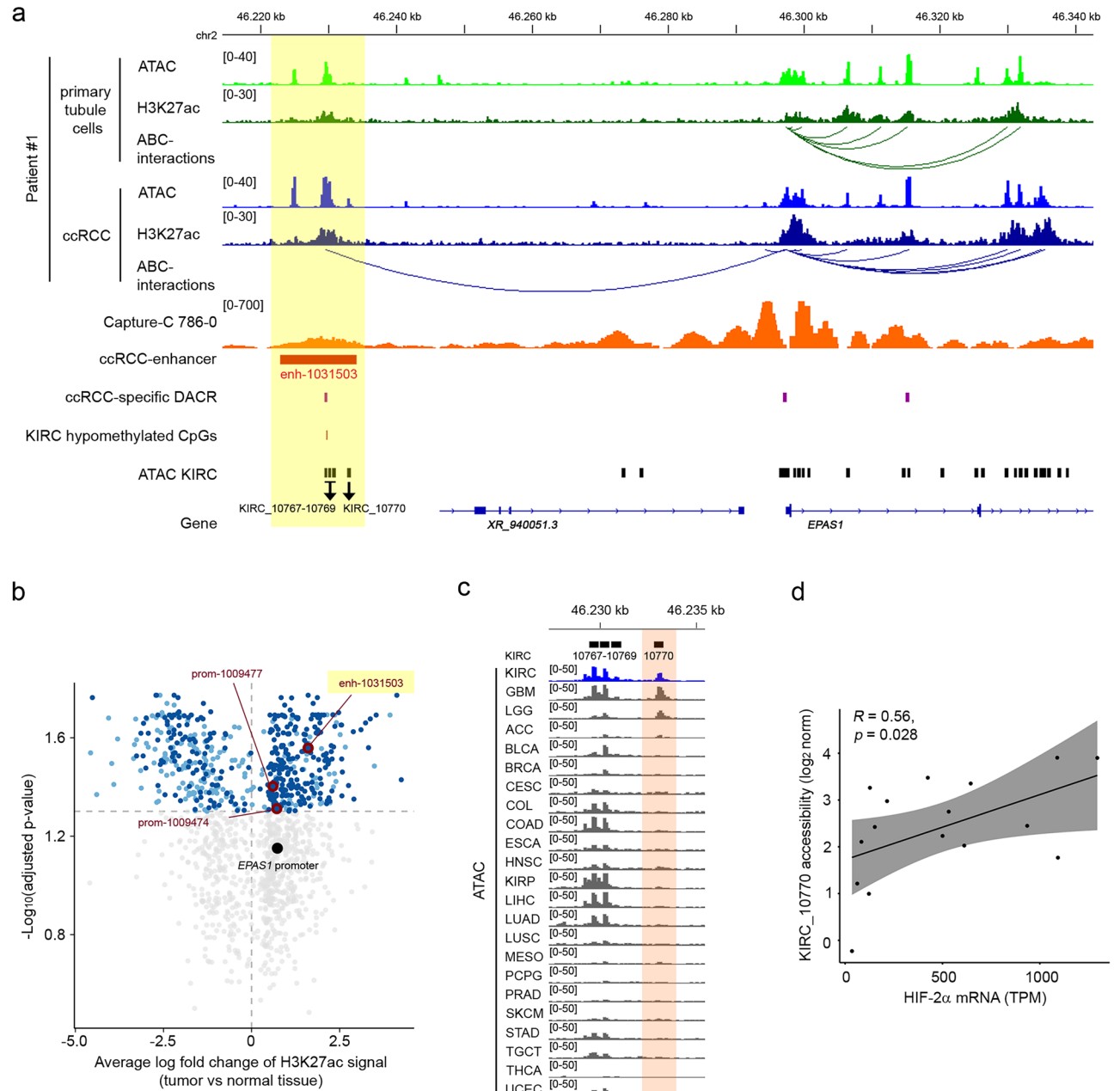

**Fig. 2 | Epigenetic analyses reveal a ccRCC-activated *EPAS1*-enhancer.**
**a** Sequencing tracks from ATAC- and H3K27ac ChIP-seq experiments performed in cells from patient #1 at the *EPAS1* locus. Activity-by-contact (ABC) analysis was used to predict enhancer-promoter interactions. Additionally, tracks from published data from Capture-C experiments in 786-0 cells[34], gained enhancer activity in ccRCC (ccRCC-enhancer)[32], ccRCC cancer-cell specific differentially accessible regions (ccRCC-specific DACR) as determined by snATAC-seq[35], KIRC hypomethylated CpGs (Beta-value cutoff 0.25, adjusted *p*-value cutoff 0.01)[36] and KIRC-specific ATAC-seq elements are shown[37]. **b** Differentially active enhancer regions on chromosome 2 in ccRCC tumors from Yao et al[32]. Average log$_2$ fold chance (H3K27ac signal, ten tumors vs corresponding normal tissue) and -log$_{10}$ adjusted *p*-value are shown. Significantly regulated elements (adjusted *p* < 0.05, light blue) that overlap

KIRC-specific ATAC sites are depicted in dark blue. Sites highlighted in red overlap KIRC-specific sites, show a positive log$_2$ fold change and are localized within 5 Mb of the *EPAS1* gene. Enhancer enh-1031503 covers elements KIRC_10767-KIRC_10770. For comparison, the H3K27ac signal at the *EPAS1* promoter is marked, but not significantly regulated. **c** ATAC-seq tracks for the different TCGA tumor entities at the ccRCC-enhancer site with gained activity in ccRCC[37]. KIRC specific ATAC-seq sites 10767-70 are indicated above the tracks. **d** Spearman's rank correlation analysis of HIF-2α mRNA expression (TPM: transcripts per million) and accessibility at KIRC_10770 in the KIRC data set. Values are from samples for which both analyses were available (*n* = 16 different tumors). Gray region depicts 95% confidence interval. Significance was assessed with a two-sided t-approximation test.

interactions overlapped between corresponding tubule and tumor cells. We measured an average of 4.8 enhancer-promoter interactions per transcription start site.

Next, we inspected the *EPAS1* locus for differential enhancer activity and enhancer-promoter interactions in the ABC data set (Fig. 2a and Supplementary Fig. 6a, Supplementary Data 4). Strikingly, this uncovered a putative regulatory loop from the *EPAS1* promoter to

an enhancer region approximately 70 kb upstream of the promoter. This loop was present in every individual-specific ABC analysis of the primary tumor cells, but absent in those from corresponding tubule cells (Fig. 2a and Supplementary Fig. 6a). The interaction was also detectable in an ABC-analysis performed with published data generated from the 786-0 ccRCC cell line (Supplementary Fig. 6b)[31–33]. To expand the epigenetic analyses of this region, we re-analyzed and

inspected published data from renal tumor cells and tumor tissue. First, an intragenic region of the *EPAS1* gene close to the promoter displayed moderate interaction signals with the putative enhancer in Capture-C experiments from a recent study exploring physical interactions of regulatory elements connected to RCC GWAS-associated polymorphisms in 786-0 cells (Fig. 2a)[34]. Second, the putative enhancer region exhibited significantly higher levels of the histone activity marker H3K27ac in the ten ccRCC tumors analyzed compared to normal tissue in a study from Yao et al. (Fig. 2a, b and Supplementary Fig. 7a, b)[32]. In these ten tumor samples, activity of this region defined by the H3K27ac signal correlated positively with *EPAS1* expression, but this correlation missed significance (R = 0.62, *p* = 0.06; Supplementary Fig. 7c). Third, the ccRCC-activated enhancer was defined as a renal cancer-specific accessible region in a recent single-cell ATAC-seq analysis (Fig. 2a)[35]. Fourth, a CpG (cg09567048) within the ccRCC-activated enhancer region was significantly hypomethylated in the analyzed KIRC cohort compared to normal tissue (Fig. 2a)[36]. Fifth, regulatory elements within the ABC-associated ccRCC-activated enhancer were defined as being predominantly open in the KIRC data set in the comprehensive catalog of accessible chromatin in tumors from the TCGA cohorts[37]. This region comprises the four regulatory elements KIRC_10767 to KIRC_10770 (Fig. 2a, c). When inspecting the ccRCC-activated enhancer in the combined tracks of the different TCGA tumors, we observed variable accessibility across tumors at KIRC_10767 to KIRC_10769, whereas at position KIRC_10770 open chromatin signals were almost exclusively restricted to KIRC tumors, albeit with some heterogeneity within the individual KIRC samples (Fig. 2c and Supplementary Fig. 8). In addition to KIRC tumors, glioblastoma (GBM) and low-grade glioblastoma (LGG) but no other tumor exhibited accessibility to KIRC_10770 (Fig. 2c). Furthermore, we compared accessibility at KIRC_10770 from the available KIRC ATAC-seq data[37] with corresponding HIF-2α mRNA expression levels (Fig. 2d). This revealed a positive correlation of accessibility of KIRC_10770 with HIF-2α mRNA expression in these tumors. Thus, higher levels of HIF-2α expression in ccRCC may be explained by an intergenic regulatory element approximately 70 kb upstream of *EPAS1* gene that is preferentially open and active in ccRCC and interacts with the *EPAS1* promoter.

## HIF-2α mRNA expression depends on pVHL and HIF

In a next step, we examined whether regulation and accessibility are dependent on the presence of pVHL. First, we reassured that augmented HIF-2α mRNA levels in the TCGA KIRC data did not correlate with copy number gains of *EPAS1*, but were associated with copy number loss of *VHL* (Fig. 3a, Supplementary Fig. 9a). Conversely, re-expression of pVHL in the VHL-defective ccRCC cell line RCC4 reduced HIF-2α mRNA levels (Fig. 3b). We also employed the RCC4/i.VHL-HA cell line which contains a doxycycline inducible *VHL* expression DNA element. Treatment of the cells with doxycycline effectively re-installed pVHL and led to clearance of HIF-α protein over time (Fig. 3c). In line with a role of pVHL in regulating *EPAS1* expression, we measured diminished HIF-2α mRNA levels in the doxycycline treated cells (Fig. 3d). This indicates that HIF-2α mRNA expression is dynamic and sensitive to the presence of pVHL in tumor cells. The best characterized function of pVHL is the recognition of hydroxylated HIF-α leading to proteasomal degradation of these HIF-subunits[6,38]. To explore a direct role of HIF in regulating *EPAS1* expression, we performed a series of experiments in ccRCC cells. Knocking-out *HIF-1β*, the HIF-α dimerization partner, in RCC4 or 786-0 cells led to reduced HIF-2α mRNA and protein levels (Fig. 3e-h). In RCC4, we also noticed an increase of HIF-1α protein upon loss of HIF-1β (Fig. 3g). This is in line with previous reports indicating an involvement of HIF in negatively regulating HIF-1α mRNA expression, possibly via induction of an antisense-RNA at the *HIF1A* locus by HIF[10,39]. In addition, we used RCC4 cells constitutively re-expressing pVHL (RCC4/VHL +) and treated

these cells with dimethyloxalylglycine (DMOG) to stabilize HIF-α. Confirming a HIF-dependent regulation, this treatment resulted in HIF-2α mRNA induction (Supplementary Fig. 10a–c).

To complement our analyses with epigenetic data, we performed ATAC-seq and H3K27ac CUT&Tag experiments in the different cell models. We noticed a moderate increase in chromatin accessibility and activity in RCC4/VHL+ cells at KIRC_10770 upon HIF-stabilization (Supplementary Fig. 10d). The same analysis in RCC4/i.VHL-HA cells treated with or without doxycycline for 18 h as well as in HIF-1β depleted or control treated RCC4 or 786-0 cells revealed that accessibility to the ccRCC-activated enhancer and especially to KIRC-10770 was strongly reduced by VHL re-expression or HIF-1β depletion, respectively (Fig. 3i). Similarly, H3K27ac levels were reduced at this site in all three cell models (Fig. 3j). From this data, we conclude that HIF regulates *EPAS1* expression in ccRCC and that KIRC_10770 is a HIF-sensitive enhancer.

## HIF interacts with the ccRCC-activated enhancer

We were interested whether the regulation of *EPAS1* expression by HIF was specific to VHL-defective cells. Thus, we proceeded to test the potential of non-transformed renal tubule cells with intact pVHL to induce HIF-2α mRNA upon HIF stabilization. For this, we used mRNA from isolated renal tubule cells derived from non-diseased tissue of nephrectomies of 105 patients and exposed these cells to the HIF-stabilizer DMOG or vehicle (Supplementary Fig. 11a). In addition to the expected increase of mRNA of the common HIF-target genes *EGLN3* and *NDRG1*, we measured a small but significant increase in HIF-2α mRNA when HIF was stabilized by DMOG (Fig. 4a, Supplementary Fig. 11b, c). In a subset of the primary tubule cells, we knocked-down HIF-1α, the main regulatory α-subunit in these cells, or HIF-1β using siRNA which abolished HIF-2α and EGLN3 mRNA induction (Fig. 4b, Supplementary Fig. 12). This result suggests that HIF regulates *EPAS1* expression in isolated renal tubule cells, albeit to a lesser extent than in ccRCC cells.

Thus, in order to explore a direct involvement of HIF via DNA-interactions in transactivating *EPAS1* expression, we re-analyzed available ChIP-seq and generated additional ChIP-seq and CUT&Tag-seq data for the different HIF-subunits from renal tubule cells and ccRCC tumor cells. In accordance with a direct role of HIF in regulating *EPAS1* expression, we discovered two sites of robust HIF-DNA interactions upstream of the *EPAS1* gene (Fig. 4c). One site overlapped an intragenic region of the *PRKCE* gene (*EPAS1*-enhancer E1) which also coincided with a hypomethylated CpG (cg08900316) in the KIRC data (Fig. 4c). At this position, HIF signals were detectable for HIF-1α and HIF-1β in primary tubule cells in which HIF-α was stabilized with DMOG, and for all three HIF-subunits in ccRCC cell lines (Fig. 4c). The second region that interacted with HIF was located within the ccRCC-activated enhancer (*EPAS1*-enhancer E2) identified by the ABC-analysis and overlapped with KIRC_10770 (Figs. 2a, 4c). Importantly, HIF signals at *EPAS1*-enhancer 2 were restricted to renal tumor cells including primary ccRCC cells when compared to primary tubule cells (Fig. 4c, Supplementary Fig. 13). We used the Cicero algorithm for testing co-accessibility of DNA-elements in single nucleus ATAC data sets to test whether *EPAS1*-enhancer E1 and E2 are linked within ccRCC tumor cells[35,40]. This revealed a positive co-accessibility score for these two elements (co-accessibility score: 0.13 between KIRC_10759 and KIRC_10770) indicating that both sites can be accessible simultaneously in the same ccRCC cell. Besides, following HIF-stabilization we also detected HIF-signals at *EPAS1*-enhancer E1 (2 out of 7 cell lines) or at E2 (2 out of 7 cell lines) in non-renal tumor cells as well as in immortalized HKC-8 renal tubule cells and VHL-competent ccRCC Caki-1 cells in data from published and newly generated sequencing experiments. This suggests that some other tumor cells may regulate *EPAS1* expression from these sites (Supplementary Fig. 14a). However, confirming the kidney cancer

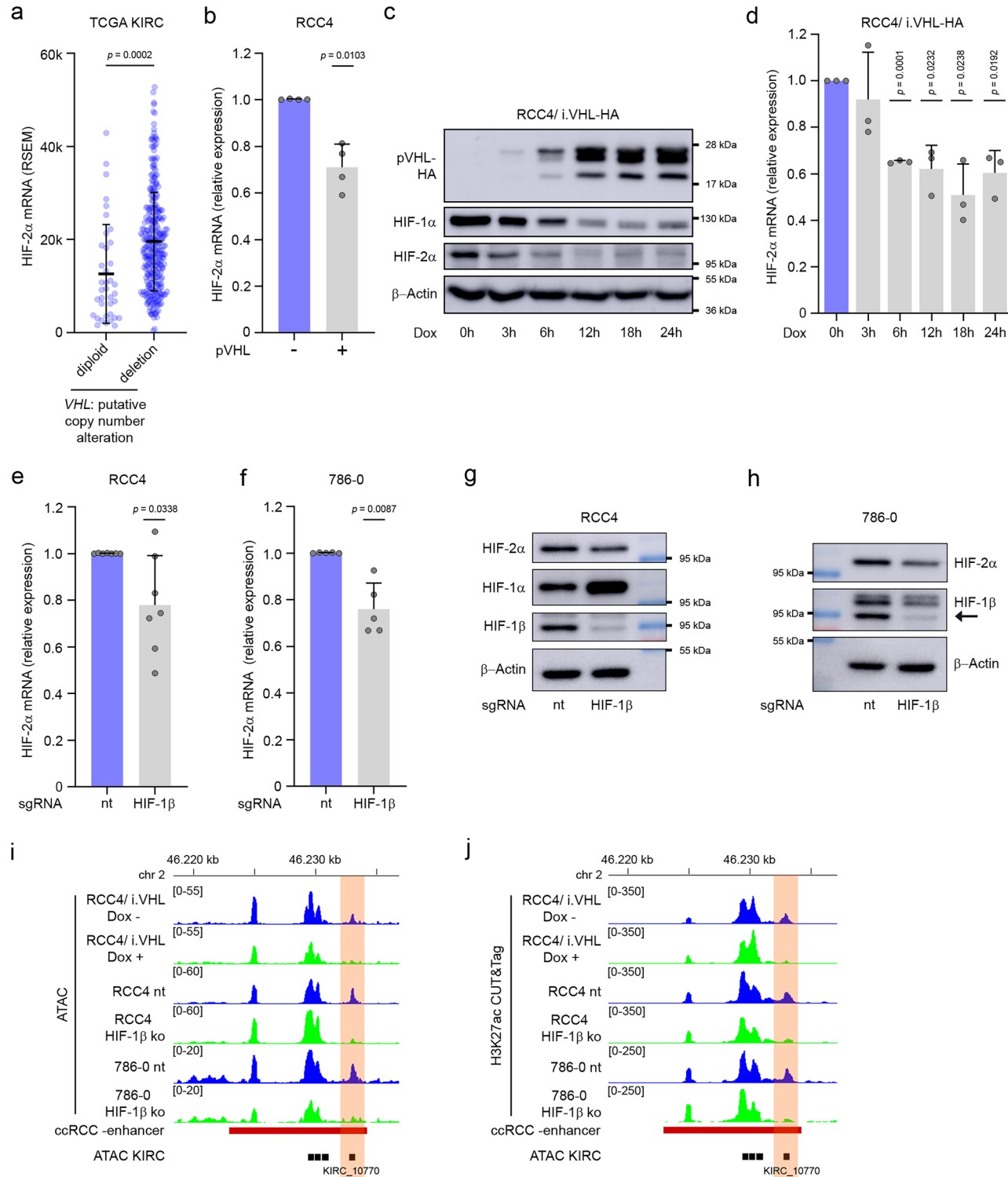

preference of this site DNAse hypersensitivity evaluated in the 733 biosamples data set from Meuleman et al[41]. was present at KIRC_10770 within *EPAS1*-enhancer E2 in samples from the renal/ cancer cluster, but not at comparable levels in other clusters (Supplementary Fig. 14b). We corroborated HIF-responsive enhancer activity for *EPAS1*-enhancer 1 and the element KIRC_10770 within *EPAS1*-enhancer 2 in reporter gene assays performed in Kelly cells exposed to DMOG or control conditions (Supplementary Fig. 14c, d, e). In order to test whether the DNA methylation status of normal or tumor tissue explains some of the binding differences for HIF at *EPAS1*-enhancer 1 and 2 between tumor and tubule cells, we resorted

to TCGA KIRC DNA methylation data[36]. Two of the three differentially methylated CpGs defined in the KIRC TCGA data set within the *PRKCE-EPAS1* locus localized in the genomic vicinity of the HIF-binding *EPAS1*-enhancers E1 and E2, respectively, further indicating their relevance for RCC biology (Supplementary Fig. 15a). However, basal methylation status in normal kidney tissue was comparable between the two CpGs and methylation was reduced at both sites in the tumors with a more pronounced effect at the CpG near *EPAS1*-enhancer 1 (Supplementary Fig. 15b). Hence, these methylation data do not explain the differences in HIF-binding signals between tubule and tumor cells.

**Fig. 3 | Expression of HIF-2α mRNA in ccRCC depends on pVHL and HIF. a** RSEM-normalized RNA-seq values for HIF-2α in the TCGA KIRC cohort stratified for *VHL* copy number alterations (diploid n = 38, deletion n = 309). *P*-values were determined by an unpaired, two-tailed t-test. **b** RT-qPCR analyses for HIF-2α mRNA in RCC4 cells with or without functional pVHL. Graph shows mean + SD. *P*-values were determined by a two-tailed one sample t-test with a hypothetical value of 1. n = 4 independent experiments. **c** Representative immunoblot (n = 2) for HIF-1α, HIF-2α, hemagglutinin (HA), and β-actin in lysates from RCC4/i.VHL-HA cells exposed to 0.5 μg/ml doxycycline to induce pVHL for the indicated time. **d** RT-qPCR analyses for HIF-2α mRNA in lysates from RCC4/i.VHL-HA cells treated with 0.5 μg/ml doxycycline for the indicated time. Expression values are mean + SD from three independent experiments. Significance was tested by a two-tailed one sample t-test with a hypothetical value of 1. **e** Expression qPCR analysis for HIF-2α mRNA in lysates from RCC4 cells transfected with sgRNA targeting HIF-1β or non-targeting (nt) control using CRISPR/Cas9. Mean + SD from 7 independent experiments.

Differences were assessed by a two-tailed one sample t-test with a hypothetical value of 1. **f** RT-qPCR analysis for HIF-2α mRNA in lysates from 786-0 cells transfected with sgRNA targeting HIF-1β or non-targeting (nt) control using CRISPR/Cas9. Graph shows mean + SD from 5 independent experiments. *P*-values were determined by a two-tailed one sample t-test with a hypothetical value of 1. **g** Representative immunoblot (n = 2) for HIF-2α, HIF-1α, HIF-1β, and β-actin from lysates of RCC4 cells transfected with sgRNA targeting HIF-1β or non-targeting (nt) control using CRISPR/Cas9. **h** Representative immunoblot (n = 2) for HIF-2α, HIF-1β, and β-actin from lysates of 786-0 cells transfected with sgRNA targeting HIF-1β or non-targeting (nt) control using CRISPR/Cas9. **i** ATAC-seq tracks at the ccRCC-activated enhancer in RCC4/i.VHL-HA cells as well as single clones of HIF-1β knockout or nt control RCC4 or 786-0 cells, respectively. RCC4/i.VHL-HA were exposed either to 0.5 μg/ml doxycycline (Dox +) or to DMSO control (Dox -) for 18 h. The ATAC element KIRC_10770 is highlighted in orange. **j** H3K27ac CUT&Tag tracks from the same cells at the same genomic region as in **i**).

As mentioned above, we detected chromatin accessibility to the ccRCC-specific HIF-binding element KIRC_10770 in the GBM and LGG TCGA ATAC-seq data (Fig. 2c)[37]. From this, we speculated that ccRCC and glioblastoma cells might regulate HIF-2α expression through a similar mechanism from a shared cancer-activated regulatory element. To explore this further, we performed ATAC-seq, HIF-1β ChIP-seq and RNA-seq experiments in U87 glioblastoma cells exposed to DMOG to stabilize HIF (Supplementary Fig. 16a). HIF-2α mRNA levels increased significantly in U87 cells after treatment with the HIF-stabilizer DMOG when measured by qPCR or RNA-seq (Supplementary Fig. 16b–d, Supplementary Data 5). Furthermore, the ccRCC-activated enhancer was accessible and interacted with HIF-1β in U87 cells indicating that glioblastoma cells share this regulatory element of HIF-2α expression with ccRCC (Supplementary Fig. 16e). Finally, similar to the results from the KIRC cohort (Fig. 2d) we measured a positive correlation of HIF-2α expression with accessibility to KIRC_10770 in the TCGA LGG data (Supplementary Fig. 16f). So far, our results indicate that the overexpression of HIF-2α mRNA in ccRCC is generated from two HIF-binding enhancers upstream of the *EPAS1* gene. *EPAS1*-enhancer 2 also displays affinity to HIF in glioblastoma.

### The *EPAS1*-enhancers interact with genetic predisposition for renal cancer

HIF-binding sites frequently coincide with renal cancer susceptibility loci indicating that genetic predisposition modulates the HIF-response in this disease[34,42,43]. In this regard, an overlap of the ccRCC-activated *EPAS1*-enhancer 2 with a HIF-2α eQTL signal for 13 polymorphisms in the Pancan eQTL catalog was observed, which was specific for the KIRC cohort and further highlights the relevance of this HIF-binding region for ccRCC biology[44] (Fig. 4c). Previous GWAS and fine-mapping analyses have revealed two independent genetic signals within intronic regions of the *EPAS1* gene associated with renal cancer development[12,21]. Of note, for one of these signals including rs12617313 a significant association with *VHL* mutations within the corresponding tumors was described for risk allele A[21]. Moreover, putative regulatory elements close to this SNP have been shown to interact with sequences within the enhancers E1 and E2 in 786-0 cells in a Capture C study (Supplementary Fig. 17A)[34]. Thus, we next explored for an interaction of HIF-dependent *EPAS1* activation and the underlying genotype for the ccRCC risk signals within *EPAS1*. To do this, we first resorted to available genotype and RNA-seq data from the KIRC cohort. We did not observe a genotype-expression correlation for SNP rs11894252 with *EPAS1* expression. rs11894252 was defined as an RCC risk SNP in the initial GWAS and was available for analysis on the genotyping array used by TCGA (Supplemental Fig. 17b)[12]. However, rs12617313, the SNP with the strongest association with *VHL*-mutations in tumors in the fine-mapping analysis, displayed a significant eQTL for *EPAS1* expression in tumor tissue, but not in corresponding normal tissue (Fig. 4d).

Of note, *EPAS1* expression levels were highest in tumors from patients homozygous for the risk allele A which is associated with *VHL*-mutations in ccRCC. We did not observe a similar genotype-expression trend for rs12617313 in the HIF-2α mRNA in tumors from the KICH and KIRP cohorts (Supplementary Fig. 17 c, d). In order to evaluate any effect of the genotype on *EPAS1* expression in primary tubule cells, we genotyped DNA from the 105 donors of primary tubule cells for rs12617313 and stratified values for basal and HIF-induced HIF-2α mRNA levels accordingly. Strikingly, similar to results from the KIRC tumors we observed strongest HIF-2α induction by DMOG in cells homozygous for the A-allele, but no difference in expression between the genotypes under control conditions (Fig. 4e). This suggests that the intronic haplotype which is associated with RCC development and *VHL*-mutations in tumors operates in a HIF-dependent manner to modulate HIF-2α mRNA expression.

### Activity of *EPAS1*-enhancer E2 depends on lineage-specific factors

De novo recruitment of HIF to enhancers in renal cancer was exemplified by comprehensive studies on a distal enhancer of *CCND1* expression[32,45]. Interestingly, HIF action at the *CCND1* enhancer was shown to be dependent on co-recruitment of the lineage-specific factor PAX8[25]. Thus, we wondered whether the renal lineage-specific factors HNF-1β and PAX8 might operate at the ccRCC-activated *EPAS1*-enhancer 2. Both transcription factors are essential for renal tumor growth[24]. We performed HNF-1β CUT&Tag-seq in 786-0 cells and inspected publicly available PAX8 ChIP-seq data from a 786-0 derived cell-line[25]. At the *EPAS1* locus, we observed DNA interactions of both factors within the HIF-interacting ccRCC-activated enhancer region, specifically KIRC_10767, suggesting that these factors are involved in the transactivation of *EPAS1* expression (Fig. 5a). In line with this hypothesis, knock-out of PAX8 or HNF-1β in 786-0 cells by CRISPR/Cas9 technology diminished HIF-2α mRNA and protein levels significantly (Fig. 5b-d). We also performed ATAC-seq and H3K27ac CUT&Tag-seq in clones of 786-0 cells with knock-out of *HNF-1*β or *PAX8*, respectively. Motifs for both transcription factors were enriched in regions with reduced accessibility when comparing knock-out to control cells confirming an epigenetic effect at binding-sites for these transcription factors (Supplementary Fig. 18, Supplementary Data 6 and 7). At the *EPAS1*-enhancer E2, both accessibility to and activity of the enhancer were substantially reduced in these knock-out clones of 786-0 cells validating the prominent role for lineage-specific transcription factors for this site (Fig. 5e). Moreover, we used one sgRNA which targeted the HNF-1β motif within KIRC_10767 as well as two sgRNA which target the DNA-sequence in the center of the PAX8 ChIP-seq peak, which did not contain a consensus PAX8 motif, in 786-0 cells (Fig. 5f, Supplementary Fig. 19). Both maneuvers introduced mutations at these sites and reduced HIF-2α mRNA as well as protein levels significantly corroborating the relevance of the specific binding-

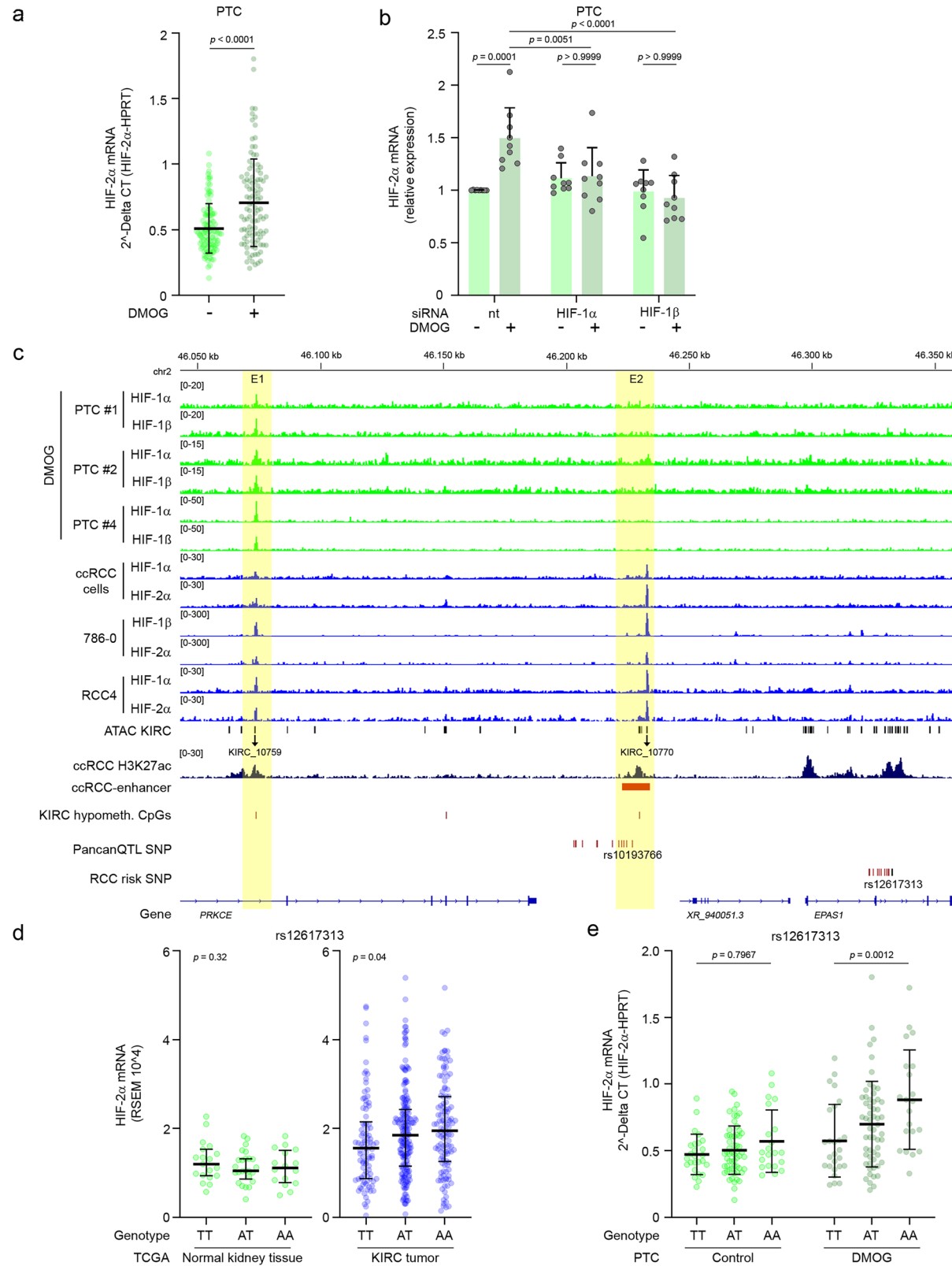

sites for *EPAS1* expression (Fig. 5g–h, Supplementary Fig. 20). We conducted HIF-1β CUT&Tag-seq in HNF-1β and PAX8 knock-out clones of cells as well as in the respective binding-site knock-out pools of cells (Supplementary Fig. 21). In all settings, we detected robust HIF-1β signals at KIRC_10770 indicating that HIF-binding to this element is not entirely depending on the presence of PAX8 or HNF-1β in the cell or at the enhancer. Taken together, the function of *EPAS1*-enhancer E2 is

governed by the interaction with renal lineage-specific transcription factors PAX8 and HNF-1β.

**Contribution of the two HIF-binding *EPAS1*-enhancers to HIF-2α expression**

Next, we wanted to confirm the function and dissect the extent of contribution of the HIF-binding elements within *EPAS1*-enhancers 1

**Fig. 4 | HIF directly interacts with *EPAS1*-enhancers. a** Expression qPCR analyses for HIF-2α mRNA in lysates from 105 isolates of primary tubule cells (PTC) exposed to 1 mM DMOG or control conditions for 16 h. Values are mean +/− SD from $n = 105$ independent experiments. Unpaired, two-tailed t-test. **b** Expression qPCR for HIF-2α mRNA in lysates from PTC depleted for HIF-1α or HIF-1β using siRNA and stimulated with or without 1 mM DMOG for 16 h. Non-targeting (nt) control siRNA was used as control. Values are mean + SD. Significance was tested by two-way ANOVA followed by Bonferroni's post hoc test. Data is from nine independent experiments using PTC from nine different individuals. **c** HIF ChIP-seq tracks from lysates of primary tubule cells (PTC, green) treated with 1 mM DMOG for 4 h and ccRCC cells (blue) at the *PRKCE-EPAS1* locus. ChIP-seq track for the activity marker H3K27ac in isolated ccRCC cells from patient #1 and hypomethylated CpGs in TCGA KIRC are also shown. KIRC-specific polymorphisms identified by the Cis-eQTL PancanQTL

analysis as an eQTL for *EPAS1* expression are indicated[44]. Germline variants associated with renal cancer development within the *EPAS1*-coding region are shown (RCC risk SNP)[12,21]. **d** Correlation of rs12617313 genotype and HIF-2α mRNA expression in the KIRC TCGA normal tissue and tumor data. Number of samples for normal tissue: Genotype TT $n = 19$, AT $n = 25$, AA $n = 16$. Number of samples for tumor tissue (KIRC): Genotype TT $n = 94$, AT $n = 188$, AA $n = 121$. Data shown are median and interquartile range. $\chi^2$-test; $p$-value as indicated for higher expression in AA individuals compared with TT individuals. **e** Genotype-expression correlation for rs12617313 and HIF-2α mRNA in PTC exposed to 1 mM DMOG for 16 h or left untreated. Number of samples for each genotype: TT $n = 26$, AT $n = 58$, AA $n = 21$. Data shown are median and interquartile range. Significance was tested by two-way ANOVA followed by Bonferroni's post hoc test.

and 2 to HIF-2α mRNA expression and to more downstream effects in renal cancer cells. To do this, we employed CRISPR/Cas9 technology targeting the enhancers in primary renal cancer cells as well as the ccRCC cell lines 786-O and RCC4. We used 2 sgRNA per enhancer region covering HIF-motifs present within the HIF-binding sites in KIRC_10759 and KIRC_10770 (Fig. 6a, Supplementary Fig. 19). First, primary cancer cells were transfected with a pool of these sgRNAs and mRNA was isolated. In line with the transactivation potential of the elements for *EPAS1* expression, we measured reduced HIF-2α mRNA levels in enhancer-knock out pools of cells validating the function of the HIF-binding elements in these explanted primary ccRCC cells (Fig. 6b). Similar results were obtained in 786-O and RCC4 cells (Fig. 6c, d). By exclusively targeting HREs in *EPAS1*-enhancer E1 effects on HIF-2α mRNA were more pronounced in 786-O cells than in RCC4 cells (Fig. 6c, d). Of note, manipulating HREs within KIRC_10770 (E2_HIF) within the ccRCC-activated *EPAS1*-enhancer E2 alone led to a striking reduction of HIF-2α mRNA by approximately 30-40% in the pools of 786-O and RCC4 cells when compared to cells transfected with control sgRNAs (Fig. 6c, d). We validated efficient knock-out of the sequences using the ICE (Interference of CRISPR Edits) tool (Supplementary Figs. 22, 23). We also cloned a selection of sequences with mutations generated by CRISP/Cas9 at both enhancers into DNA vectors and performed luciferase reporter assays. Confirming a functional role for the HREs in binding HIF and activating *EPAS1* transcription, sequences with mutations led to a substantial reduction of luciferase activity in DMOG-stimulated Kelly cells when compared to the wild-type sequence (Supplementary Fig. 24). Importantly, reduction of HIF-2α mRNA also translated into reduced expression of HIF-2α protein in 786-O and RCC4 cells underlining the relevance of the enhancers for the HIF-2 signaling pathway (Fig. 6e, f). We focused on the ccRCC-activated *EPAS1*-enhancer 2 and performed RNA-seq from lysates of *EPAS1*-enhancer 2 HRE-defective (E2_HIF) and control 786-O and RCC4 cell populations. Analysis for differentially expressed genes revealed 18 significantly upregulated and 32 downregulated genes for 786-O and 39 upregulated and 64 downregulated genes for RCC4 cells (log₂ fold change: >0.5/< −0.5, adjusted $p < 0.05$, Supplementary Data 8 and 9). Pathway analysis using the MsigDB "Hallmark" gene set confirmed significant enrichment of the hypoxia pathway in mRNA transcripts which were downregulated in the enhancer knock-out 786-O or RCC4 cell pools, respectively, when compared to control cells (Fig. 6g, h). Analyzing the overlap of significantly regulated genes in *EPAS1*-enhancer E2 knock-out 786-O and RCC4 cells revealed a concordance in gene regulation between the two cell lines including reduced expression of CCND1, NDRG1 and VEGFA transcripts (Fig. 6i).

### Relevance of the ccRCC-activated *EPAS1*-enhancer 2 for downstream targets and tumor biology

In the following step, we generated *EPAS1*-enhancer 2 HRE-defective (E2_HIF) clones of 786-O cells to explore the effect size to which this enhancer contributes to HIF-2 dependent tumor biology and

transcriptional regulation. E2_HIF mutated clones of cells had reduced HIF-2α protein levels compared to control clones of cells (Fig. 7a). Importantly, inactivation of *EPAS1*-enhancer 2 (E2_HIF) prevented tumor formation of 786-O cells in a tumor xenograft mouse model when compared to control cells, indicating an essential effect of this enhancer for ccRCC growth (Fig. 7b, c, Supplementary Fig. 25a). Our results are in line with a recent CRISPR inhibitor screen examining the function of HIF-associated enhancers in ccRCC which described reduction of tumor growth by targeting the ccRCC-activated *EPAS1*-enhancer 2 with sgRNAs also employed in our study (Supplementary Fig. 25b)[25].

To follow the effects from the tumor model up, we evaluated the downstream transcriptional effects of the *EPAS1*-enhancer 2 (E2_HIF). We performed RNA-seq from lysates of two control (nt) and two *EPAS1*-enhancer 2 HRE-defective (E2_HIF) clones of 786-O cells. In a differential expression analysis, downregulation of the HIF-2 target genes *NDRG1*, *VEGFA*, and *CCND1* in *EPAS1*-enhancer 2 HRE-defective (E2_HIF) clones of 786-O cells confirmed the critical function of this enhancer for downstream transcriptional effects of HIF-2α (Fig. 7d, Supplementary Data 10).

In order to define the direct HIF-2 component of the response, we intersected the RNA-seq results with available HIF ChIP-seq from 786-O cells using gene set enrichment analysis[45]. This confirmed a significant enrichment of 786-O HIF-binding sites amongst genes being downregulated in *EPAS1*-enhancer 2 HRE-defective (E2_HIF) cells (Fig. 7e). We further applied CUT&Tag technology for HIF-1β to interrogate disturbances of HIF-DNA interactions at the genome level in *EPAS1*-enhancer 2 HRE-defective (E2_HIF) clones of 786-O cells. As expected, we detected absent HIF-1β protein at KIRC_10770 in the *EPAS1*-enhancer 2 HRE-defective (E2_HIF) cells, but reasonably preserved interactions at a positive control site within a known *EGLN3* HIF-binding enhancer (Fig. 7f). In the knock-out clones of cells, ATAC-seq and H3K27ac CUT&Tag-seq experiments revealed reduced accessibility and activity at KIRC_10770 (Fig. 7f). Averaging CUT&Tag signals for HIF-1β and H3K27ac from *EPAS1*-enhancer 2 HRE-defective (E2_HIF) clones of 786-O cells across the set of ChIP-seq defined HIF-binding sites centered on the respective H3K27ac signal revealed reduced HIF-1β and H3K27ac signals compared to control cells (Fig. 7g). Signals for HIF-1β and H3K27ac at non-HIF-binding enhancers were comparable between the two cell populations. This shows that downstream HIF-DNA interactions at and activity of HIF-binding enhancers depend on an intact *EPAS1*-enhancer E2.

Supporting the HIF-2α-specific effect and highlighting the relevance of the enhancer for renal cancer biology, the transcriptional response upon targeting the *EPAS1*-enhancer 2 overlapped strikingly with effects generated by treatment of 786-O cells with the specific HIF-2 inhibitor PT2385[46] (Fig. 7h).

In summary, distal enhancers of *EPAS1*-expression interact directly with HIF in ccRCC. Together with lineage-specific factors HNF-1β and PAX8, they form a transcriptional autoregulatory circuit of the HIF-2 signaling pathway activation in these tumors which is modulated

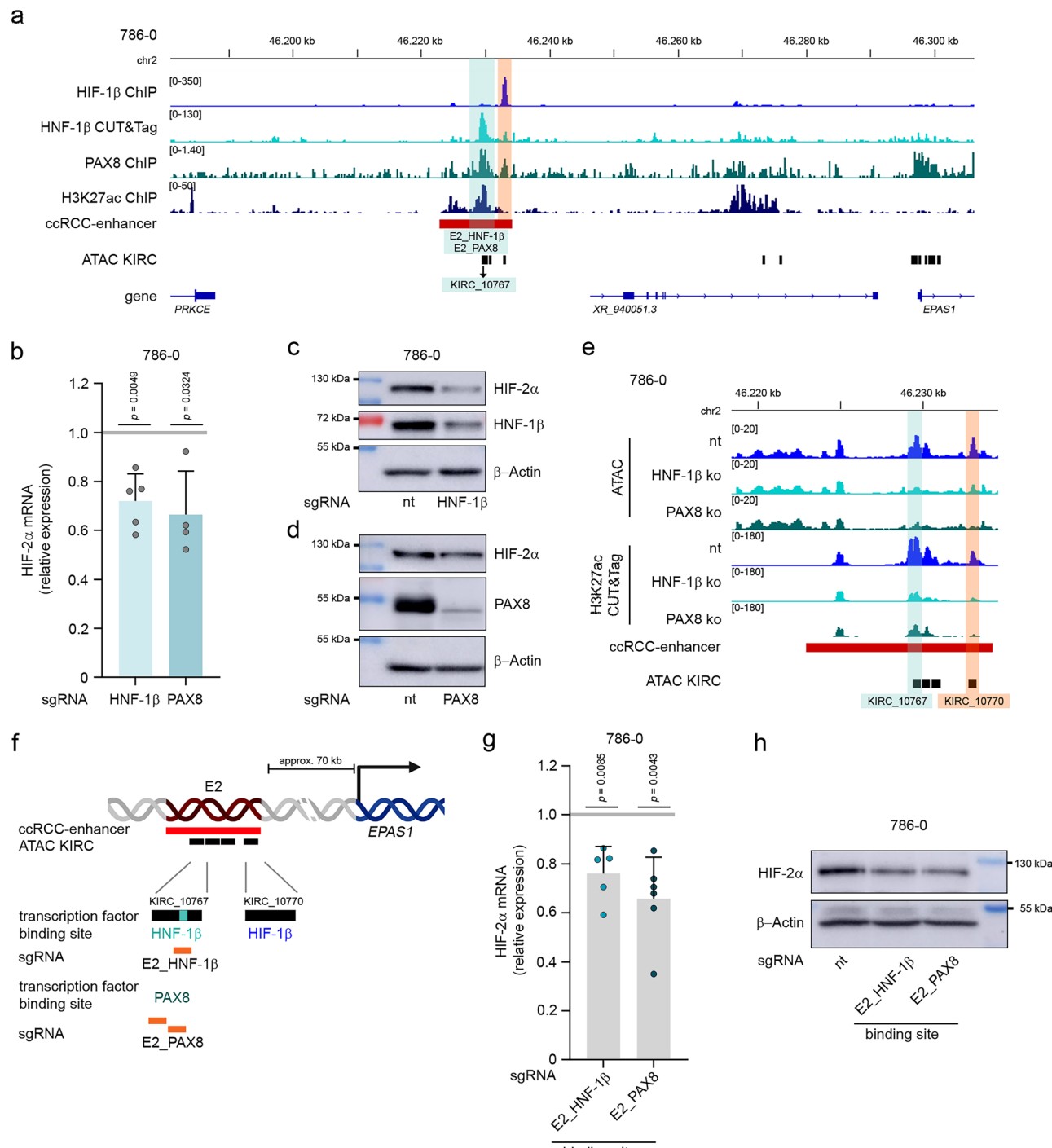

**Fig. 5 | Lineage-specific transcription factors govern activity of *EPAS1*-enhancer E2. a** ChIP-seq tracks for HIF-1β, PAX8[25], H3K27ac and HNF-1β CUT&Tag-seq track from 786-0 cells at *EPAS1*-enhancer E2. Binding sites for HNF-1β (E2_HNF-1β) and PAX8 (E2_PAX8) within KIRC_10767 are highlighted in blue. HIF-binding site is marked in orange. **b** Expression qPCR analyses for HIF-2α mRNA in lysates from 786-0 cells depleted for the indicated transcription factor (HNF-1β or PAX8). Values are mean + SD from 5 (HNF-1β) or 4 (PAX8) independent experiments. Statistical significance was assessed by a two-tailed one sample t-test with a hypothetical value of 1. **c** Immunoblot analyses for HIF-2α, HNF-1β, and β-actin in lysates from 786-0 cells depleted for HNF-1β using CRISPR/Cas9. Representative blot from two independent experiments with similar results. **d** Immunoblot analyses for HIF-2α, PAX8, and β-actin in lysates from 786-0 cells depleted for PAX8 using CRISPR/Cas9. Representative blot from two independent experiments with similar results. **e** ATAC-seq and CUT&Tag-seq tracks of the chromatin activity marker H3K27ac from single clones of HNF-1β knock-out, PAX8 knock-out or control (nt) clones of

786-0 cells at *EPAS1*-enhancer E2. The ATAC KIRC element 10767 harboring binding sites for HNF-1β and PAX8 is highlighted in blue, the HIF-1β binding site at KIRC_10770 is marked in orange. **f** Schematic depiction of transcription factor binding sites for HNF-1β, PAX8 and HIF-1β at *EPAS1*-enhancer E2. Positions of sgRNAs used for knock-out of the respective binding sites are indicated in orange. Created in BioRender. Naas, S. (2025) https://BioRender.com/w24j560. **g** Expression qPCR analysis for HIF-2α mRNA in pools of 786-0 cells transfected with sgRNA targeting the HNF-1β (E2_HNF-1β) or PAX8 (E2_PAX8) binding site at the *EPAS1*-enhancer E2. Cells treated with non-targeting sgRNA served as controls. Values are mean + SD. *p*-values were determined by a two tailed one sample t-test with a hypothetical value of 1. *n* = 5 (E2_HNF-1β) or *n* = 6 (E2_PAX8) independent experiments. **h** Immunoblot analyses for HIF-2α and β-actin in 786-0 cells with intact (nt) or mutated binding sites for HNF-1β (E2_HNF-1β) or PAX8 (E2_PAX8) at the *EPAS1*-enhancer E2. Representative blot from two independent experiments with similar results.

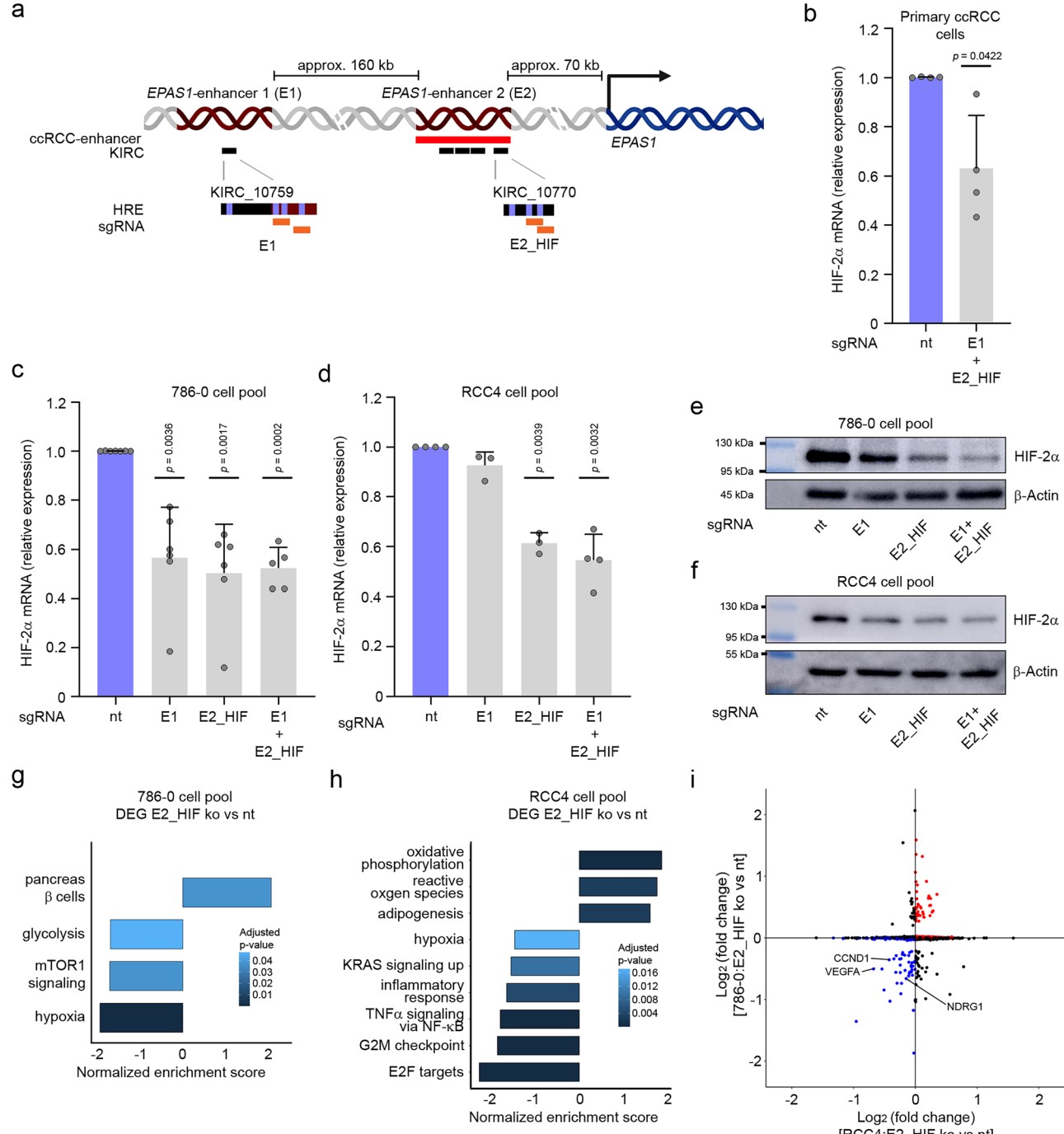

**Fig. 6 | Targeting HIF-motifs within *EPAS1*-enhancers E1 or E2 affects transcription of HIF-2α. a** Schematic of the two HIF-binding *EPAS1*-enhancers E1 and E2 at the *PRKCE-EPAS1* locus. The location of HIF-binding motifs and positions of sgRNAs within the KIRC elements are indicated. Created in BioRender. Naas, S. (2025) https://BioRender.com/w24j560. **b** Expression qPCR for HIF-2α mRNA in lysates from primary ccRCC cells isolated from 4 different tumor nephrectomies transfected with a pool of sgRNAs directed against the HIF-binding-sites at *EPAS1*-enhancer 1 (E1) and *EPAS1*-enhancer 2 (E2_HIF) or a control sgRNA (nt). Values are mean + SD from 4 independent experiments. Significance was determined by a two tailed one sample t-test with a hypothetical value of 1. Expression qPCR for HIF-2α mRNA in lysates from 786-0 (**c**) or RCC4 (**d**) cells transfected with sgRNAs targeting *EPAS1*-enhancer 1 (E1), the HIF-binding site of *EPAS1*-enhancer 2 (E2_HIF) or both enhancers (E1 + E2_HIF). nt = non-targeting control sgRNA. Values are mean + SD from independent experiments (786-0: n = 7 for nt; *n* = 6 for E1 and E2_HIF; *n* = 5 for

E1 + E2_HIF; RCC4: *n* = 4 for nt and E1 + E2 HIF; *n* = 3 for E1 and E2_HIF). Significance was tested by a two tailed one sample t-test with a hypothetical value of 1. Immunoblot analyses for HIF-2α and β-actin protein in lysates from 786-0 (**e**) or RCC4 (**f**) cells transfected with sgRNAs targeting the enhancers as indicated. Representative blot from three (786-0) or two (RCC4) independent experiments with similar results. Gene set enrichment analysis for differentially expressed genes comparing RNA-seq data generated in 786-0 (**g**) or RCC4 (**h**) cells transfected with sgRNA targeting E2_HIF or non-targeting control guides (nt). One RNA sample of each condition was sequenced in technical duplicates. **i** Comparison of log₂ (fold change) expression values of genes significantly regulated upon E2_HIF knock-out in 786-0 or RCC4 cells. HIF-target transcripts NDRG1, VEGFA, and CCND1 are indicated. One RNA sample of each condition was sequenced in technical duplicates.

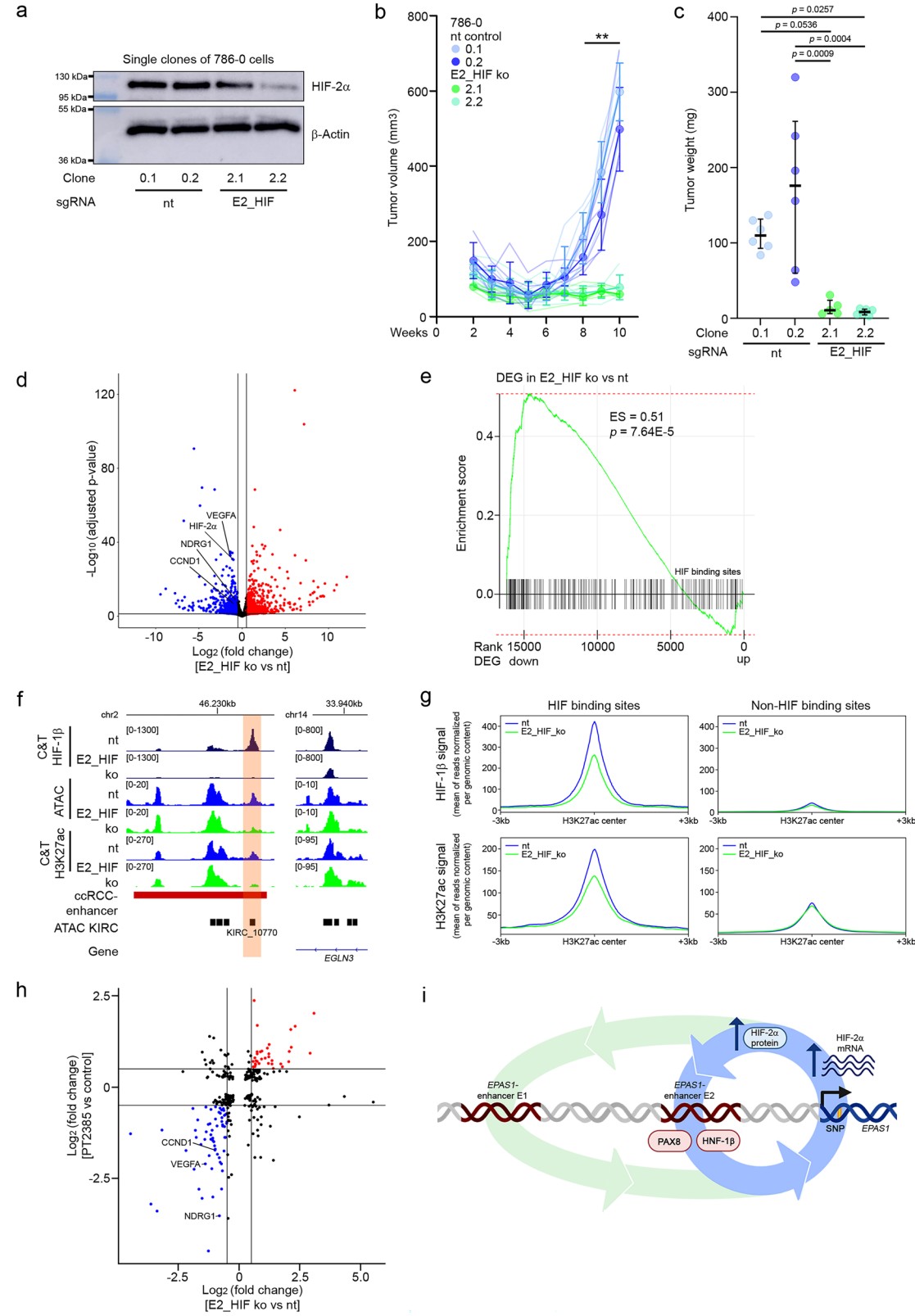

by RCC-associated genetic variation and is indispensable for tumor growth (Fig. 7i).

## Discussion

Our work defines a transcriptional regulatory circuit of HIF-2α expression in renal cancer[47]. This loop comprises the recruitment and activation of distal enhancers of *EPAS1* which respond to HIF itself and additionally relies on the presence of the renal lineage-specific factors PAX8 and HNF-1β explaining some of the extraordinary tissue-specificity of the HIF-2 response. Moreover, we provide evidence that the circuitry described above interferes with genetic predisposition signals within the *EPAS1* gene, for which the risk haplotype is tightly connected to RCC development, *VHL* alterations as well as enhanced *EPAS1* expression.

**Fig. 7 | EPAS1-enhancer 2 controls downstream effects of HIF-2α and tumor growth. a** Representative immunoblot (*n* = 3) for HIF-2α and β-actin protein in lysates from single clones of *EPAS1*-enhancer 2 defective (E2_HIF) or control (non-targeting, nt) 786-0 cells. **b** Xenograft tumor assay in NOD/SCID-gamma mice using *EPAS1*-enhancer 2 defective (E2_HIF ko) or intact (nt) clones of 786-0 cells. Volume values are mean ± SD from 6 tumors per cell clone. Two-way ANOVA followed by Bonferroni's post hoc test comparing tumor volume between mean nt and E2_HIF ko at the indicated time points. Significant differences (**p < 0.01) were detected for comparisons between mean values of E2_HIF ko (2.1 or 2.2) with nt (0.1 or 0.2) xenografts for weeks 8, 9 and 10. **c** Weight of the explanted xenograft tumors. Data shown are median and interquartile range from 5 (E2_HIF 2.1) or 6 (nt 0.1, nt 0.2, E2_HIF2.2) tumors. One-way ANOVA followed by Bonferroni's post hoc test. **d** Volcano plot from RNA-seq experiments in E2_HIF ko versus nt clones of 786-0 cells. HIF-2α mRNA as well as HIF-target transcripts NDRG1, VEGFA, and CCND1 are indicated. Data is from the two clones of cells per condition from a) and b). **e** GSEA for HIF-binding sites in 786-0 cells and RNA-seq data (*n* = 2 clones from d). DEG: differentially expressed genes, ES: enrichment score. *p*-value was estimated using adaptive multilevel Monte Carlo scheme and adjusted for multiple testing with the Benjamini–Hochberg method. **f** CUT&Tag-seq (C&T) for HIF-1β and H3K27ac as well as ATAC-seq at *EPAS1*-enhancer 2 and the control locus *EGLN3* from E2_HIF ko versus nt clones of 786-0 cells. **g** CUT&Tag-seq signals for HIF-1β or H3K27ac integrated over 543 ChIP-seq pre-defined HIF1-β binding sites in E2_HIF ko versus nt clones of 786-0 cells. Non-HIF binding enhancers served as control. Signals were centered on the H3K27ac peak within the enhancers. **h** Comparison of log$_2$ (fold change) expression values of genes significantly regulated in E2_HIF ko 786-0 cells (*n* = 2 clones) or through HIF-2 inhibition by PT2385 in 786-0 cells[46]. HIF-target transcripts NDRG1, VEGFA, and CCND1 are indicated. **i** The transcriptional circuitry of HIF-2α expression at the *EPAS1* gene locus. Created in BioRender. Naas, S. (2025) https://BioRender.com/w24j560.

The *EPAS1*-enhancer 2 appears to be preferentially active in ccRCC which is in line with our previous observations that HIF occupies important cell type-specific oncogenic enhancers in ccRCC, e.g., for *CCND1* and *MYC* expression[43,45]. Thus, we provide evidence that a large proportion of VHL-deficient renal tumors establishes an *EPAS1*-associated transcriptional regulatory circuitry that allows for continuous promotion of *EPAS1* expression. The *EPAS1*-enhancer 2 with its ccRCC-specific HIF-binding element KIRC_10770 is central to this regulatory mechanism. Although other renal tumors displayed broad accessibility to the more upstream part of the enhancer (e.g., KIRP in Fig. 2c), they did neither show accessibility to the HIF-binding site nor upregulated HIF-2α mRNA levels. In ccRCC cells, any disturbance within the *EPAS1*-enhancer 2 (e.g., genetically) or within composition of the interacting transcription factors (e.g., inhibition of HIF-2, loss of HIF-2, loss of the lineage-specific factors HNF-1β or PAX8) reduced *EPAS1* expression as well as the downstream transcriptional output of HIF-2. This effect is detrimental for cancer cells depending on a functional circuitry that maintains oncogenic signaling. Our work and the recent reports on lineage-specific factors would strongly support this hypothesis[24,25].

In ccRCC, HIF-DNA binding significantly coincides with genetic loci for renal cancer predisposition[34,42,48]. For example, over-expression of *CCND1* by HIF specifically in ccRCC is mediated via a distal transcriptional enhancer which overlaps with one of the strongest signals for RCC development[10,25,45,49]. HIF-binding to this enhancer is modified by the underlying genotype[45]. Although, the HIF-binding sites at the *EPAS1* locus do not directly interfere with the intronic RCC risk signals, the enhancers are physically linked to regulatory DNA sequences close to the risk SNPs within the gene[34]. Furthermore, rs12617313 is an eQTL for *EPAS1* expression in tumors as well as in primary tubule cells in which HIF was stabilized. rs12617313 has been defined as an RCC risk SNP in a fine-mapping study comprising a 120 kb region including *EPAS1*[21]. The risk allele A showed the strongest association with genetic *VHL* alterations within tumors. In the KIRC cohort, HIF-2α mRNA levels were highest in tumors from patients homozygous for the A-allele. Similarly, HIF-stabilization in primary tubule cells triggered a genotype-expression correlation for HIF-2α mRNA with highest levels in cells from individuals carrying two A-alleles. So far, we could not determine the precise mechanism how this differential regulation is mediated by the polymorphisms, but our results provide a plausible explanation for why these SNPs predispose to RCC and associate with genetic *VHL* alterations. Since genotype-dependent HIF-2α expression was also present in non-tumor tubule cells upon HIF-stabilization, one would expect that loss of functional pVHL together with HIF-stabilization would allow this regulation to become effective early on in ccRCC development. In this respect, HIF-2α protein was shown to be overexpressed in early lesions in kidney of VHL patients[19].

In the United States and Europe, the HIF-2 inhibitor belzutifan has been licensed for the treatment of VHL-associated tumors in VHL-patients and of advanced sporadic RCC. Belzutifan in combination with established RCC therapeutics met primary endpoints in phase 2 and 3 clinical trials in patients with advanced sporadic RCC[14–16]. However, inhibition of HIF-2 was ineffective in reducing cell growth in a number of ccRCC cell lines[18]. Moreover, some renal tumors appeared to be resistant to HIF-2 inhibition suggesting that a certain fraction of ccRCC tumor cells do not rely on an intact HIF-2 signaling pathway. In fact, in experimental models some tumors developed resistance to belzutifan through mutations in *EPAS1* or *HIF-1β*[17]. In line with this heterogeneity, we observed that *EPAS1*-enhancer 2 accessibility correlates with HIF-2α mRNA levels in ccRCC tumors and that in some of the tumors the HIF-binding site was not accessible. Importantly, tumor xenografts with high levels of HIF-2α mRNA and protein were much more sensitive to inhibition than those with lower levels as described in the study by Chen et al.[17]. Thus, measuring activity of the ccRCC-activated *EPAS1*-enhancer 2 may provide direct and helpful information about susceptibility of ccRCC tumor cells to HIF-2 inhibition. Moreover, given its high tissue-specificity the HIF-interacting element KIRC_10770 may represent a promising target to reduce HIF-2 activity in belzutifan resistant tumors with acquired mutations in HIF-2α. The potential of this site as a target is further supported by our findings that reducing activity of the *EPAS1*-enhancer 2 lowered expression of downstream HIF-2 target genes relevant to ccRCC biology and therapy (such as *CCND1* and *VEGFA)* and most importantly prevented xenograft growth.

Our results suggest the existence of a similar HIF-2α regulatory loop in glioblastoma. In fact, previous reports highlighted activation of the HIF pathway in glioblastoma[50]. Specifically, HIF-2α mRNA expression was upregulated by hypoxia in glioma stem cells. We defined the molecular mechanism underlying this finding indicating that potential responses of brain tumors to HIF-2 inhibition could depend on the presence of this HIF-2 regulatory circuitry.

Taken together, our data provide evidence for the recruitment and maintenance of a transcriptional regulatory circuitry of *EPAS1* expression in renal clear cell cancer which is dependent on HIF itself and lineage-specific transcription factors. The activity of this circuitry is modulated by genetic predisposition and differences in the magnitude of its output may explain heterogeneity in responses to anti-HIF-2 therapy in renal cancer patients.

## Methods

### Healthy kidney and tumor tissue samples

Tumor and healthy human kidney cortical tissue from anonymized donors undergoing tumor nephrectomy was provided by the Comprehensive Cancer Center Erlangen-EMN (CCC) at the Uniklinikum Erlangen. Each patient gave written informed consent and the local ethics committee at the Universität Erlangen-Nürnberg approved use of the tissue (329_16B; 542_20Bc). We did not conduct sex- or gender-specific analyses, as this was not the focus of the study. Specimens were collected in accordance with the *World Medical Association*

*Declaration of Helsinki*. Tumor and normal kidney samples were examined and diagnosed by an expert pathologist. From our Erlangen RCC cohort we used tumor tissue and corresponding normal kidney tissue from 114 clear cell, 16 papillary and 11 chromophobe renal cancer patients for RNA analysis.

### In situ hybridization using RNAscope

In situ hybridization experiments were performed on formalin-fixed paraffin-embedded kidney tissue using the RNAscope 2.5 HD Assay - BROWN (322300, RNAscope 2.5 HD Reagent Kit, Advanced Cell Diagnostics, Bio-Techne GmbH Wiesbaden, Germany) according to the manufacturer's instructions. In short, paraffin blocks were cut in 5 μm sections using a microtome and mounted on Superfrost plus slides. After air drying slides overnight, slides were baked for 1 h at 65 °C and deparaffinized using xylene and 100% ethanol. Further pretreatment included incubation with target retrieval reagents, hydrogen peroxide and protease plus reagents. HIF-2α target mRNA was hybridized with the RNAscope Probe Hs-EPAS1 (410591, Advanced Cell Diagnostics, Bio-Techne GmbH Wiesbaden, Germany) which has been described earlier[51]. Six amplification steps were performed before signal detection using DAB. Slices were counterstained with 50% Hematoxylin and mounted with BioCare EcoMount (320409, Biocare Medical, Pacheco, CA, USA). Positive and negative control probes were routinely included to ensure interpretable results.

Co-staining for CA9 (polyclonal rabbit, NB100-417, Novus Biologicals, Centennial, CO, USA; dilution 1:500) and CD31 (monoclonal mouse, #3528, Cell Signaling, Danvers, MA, USA; dilution: 1:200) was performed on RNAscope-preanalyzed sections, which were cooked for 12 min in 1X Target Retrieval Solution (pH 6.1, S169984-2, Agilent Dako, Santa Clara, CA, USA), washed with PBS, and incubated in blocking solution (5% horse serum in PBS) for 1 h. Subsequently, sections were washed in PBS and incubated with primary antibodies overnight followed by thorough washing with PBS. As secondary antibodies goat anti-mouse IgG (Alexa Fluor 555, A-21422, Thermo Scientific, Waltham, MA, USA; dilution: 1:500) or donkey anti-rabbit (Alexa Fluor 488, A-21206, Thermo Scientific, Waltham, MA, USA; dilution: 1:500) were used. After incubation in secondary antibody solution for 60 min, sections were washed in PBS and mounted. Staining was visualized with a Leica DM 6000B microscope (Leica, Wetzlar, Germany).

### Cell culture

786-0 and Caki-1 cells were purchased from ATCC. Kelly cells were a gift from C. Warnecke and U-87 cells were a gift from F. Müller, Erlangen, Germany. RCC4 and RCC4/i-VHL-HA cells were from Sir P. Ratcliffe, Oxford, UK. Cell lines were regularly tested for mycoplasma infection and authenticated before being used. 786-0, RCC4, and U-87 cell lines were grown in Dulbecco's modified Eagle's Medium supplemented with 100 U/ml penicillin, 100 μg/ml streptomycin and 10% fetal bovine serum (Sigma Aldrich, St. Louis, MO, USA). Kelly cells were maintained in RPMI 1640 medium supplemented with 100 U/ml penicillin, 100 μg/ml streptomycin and 10% fetal bovine serum. Caki-1 cells were grown in McCoy´s 5 A Medium (Gibco, Thermo Fisher Scientific, Waltham, MA, USA) supplemented with 100 U/ml penicillin, 100 μg/ml streptomycin and 10% fetal bovine serum (Sigma Aldrich, St. Louis, MO, USA). Primary cell isolation from healthy human kidney cortical tissue or tumor tissue from patients undergoing tumor nephrectomy was performed as described previously[52]. Samples were minced, digested with collagenase II (Gibco, Thermo Fisher Scientific, Waltham, MA, USA), and sieved through 100 μm and subsequently 70 μm filters. Primary human tubule cell cultures were maintained in Dulbecco's modified Eagle's medium/Ham's F-12 supplemented with 2.5% (day 1) or 0% (day 2 onwards) fetal calf serum, 2 mM L-glutamine, 100 U/ml penicillin and 100 μg/ml streptomycin, 5 μg/ml insulin, 5 μg/ml transferrin, and 5 ng/ml selenium (Sigma Aldrich, St. Louis, MO, USA), 10 ng/ml triiodothyronine (T3, Sigma Aldrich, St. Louis, MO, USA), 1 mg/ml hydrocortisone and 100 μg/ml epidermal growth factor (Peprotech, Hamburg, Germany). Epithelial origin was confirmed by immunocytochemistry for N- and E-Cadherin. Sub-confluent cell cultures were exposed to 1 mM dimethyloxalylglycine (DMOG, Cayman Chemicals, Ann Arbor, MI, USA) as indicated to induce HIF-α stabilization. Primary ccRCC tumor cell cultures were maintained in Dulbecco's modified Eagle's Medium supplemented with 100 U/ml penicillin, 100 μg/ml streptomycin and 10% fetal bovine serum. RCC4/i-VHL-HA cells were treated with to 0.5 μg/ml doxycycline (D3447, Sigma Aldrich, St. Louis, MO, USA) for the indicated time to induce VHL expression.

### siRNA transfection

siRNAs targeting HIF-1α and HIF-1β subunits and control non-targeting siRNA against drosophila HIF (nt) have been previously described[53]. siRNAs were transfected at a final concentration of 40 nM using Lipofectamine 3000 (ThermoFisher Scientific, Waltham, MA, USA) transfection reagent according to the manufacturers' instructions.

### DNA isolation and genotyping

Genomic DNA was isolated by phenol/chloroform extraction from tubule cells. We used a TaqMan® assay (C_26434874_10, 4351379 and TaqMan Genotyping-Master Mix: 4371355, Thermo Fisher Scientific, Waltham, MA, USA) to genotype for rs12617313. Data were analyzed using the TaqMan® Genotyper Software V1.3 (Thermo Fisher Scientific, Waltham, MA, USA).

### Genome editing

For genome editing of primary ccRCC tumor cells, 786-0 and RCC4 cells sgRNA directed against exonic regions of HIF-1β, HNF-1β, PAX8 or enhancer regions (HIF-1β, HNF-1β and PAX8 binding sites) at the *PRKCE-EPAS1* locus were designed according to algorithms provided by the Zhang lab using CRISPOR information in the UCSC Genome Browser[54,55] (Supplementary Data 11). The two sgRNAs targeting HREs within *EPAS1*-enhancer-2 corresponded to two sgRNAs used by Patel et al. for the CRISPRi screen[25]. Knock-out of the region of interest was performed by transfecting TrueCut Cas9 Protein (Thermo Fisher Scientific, Waltham, MA, USA) together with targeting sgRNAs (A35533, Trueguide Synthetic sgRNA, Thermo Fisher Scientific, Waltham, MA, USA) or non-targeting control sgRNAs (A35526, TrueGuide sgRNA Negative Control, non-targeting, Thermo Fisher Scientific, Waltham, MA, USA) using electroporation. Single clones of cells were generated by dilution. For mutation screens, genomic DNA of each clone was isolated by phenol-chloroform extraction. The region targeted by CRISPR/Cas9 was amplified by PCR and analyzed by Sanger sequencing. Knock-out of enhancer regions in cell pools was evaluated using the "Inference of Crispr Editing" method (ICE v2 CRISPR Analysis Tool, Synthego, Menlo Park, CA, USA). Single clones of cells were evaluated for effective knock-out of regulatory sites by cloning the respective regions in a pGL3 promoter vector (Promega, Madison, WI, USA) and subsequently performing Sanger sequencing. For *EPAS1* enhancer knock-out experiments off-target copy number alterations of the *EPAS1* gene was excluded by copy number determination. For sequences of sgRNAs used in this work please see Supplementary Data 11.

### RNA isolation and quantitative RT−PCR

RNA from cells or fresh frozen tissue was isolated using peqGold total RNA kit (VWR Peqlab, Erlangen, Germany) according to the manufacturer´s protocol and transcribed into cDNA using the high capacity cDNA reverse transcription kit (4368814, Thermo Fisher Scientific, Waltham, MA, USA). Quantitative RT−PCRs were performed using Maxima SYBR Green/ROX qRT-PCR Master Mix (Ko233, Thermo Fisher Scientific, Waltham, MA, USA) on a StepOnePlus Real-Time PCR cycler (Applied Biosystems, ThermoFisher Scientific, Waltham, MA, USA). Expression values were normalized to values for the housekeeping gene *HPRT*. For expression primers please see Supplementary Data 11.

## Western blotting

Cells were lysed in UREA/SDS buffer. Proteins were resolved by SDS-PAGE and transferred onto PVDF membranes. Rabbit polyclonal anti-HIF-1α antibody (Cay10006421, rabbit polyclonal, Cayman Chemicals, Ann Arbor, MI, USA; dilution: 1:1,000) and monoclonal mouse anti-HIF-1α antibody (610959, BD Biosciences, New Jersey, USA; dilution: 1:500), goat polyclonal anti-HIF-2α antibody (AF2997, R&D Systems, Minneapolis, USA; dilution: 1:250) and rabbit monoclonal anti-HIF-2α antibody (A700-003, Bethyl, Waltham, USA; dilution: 1:1,000) as well as rabbit polyclonal anti-HIF-1β antibody (NB100-110, Novus Biologicals, Biotechne, Wiesbaden, Germany; dilution: 1:1,000), monoclonal rat anti-hemagglutinin (HA) antibody (11867423001, Roche Diagnostics GmbH, Mannheim, Germany; dilution: 1:1,000) and polyclonal rabbit anti-VHL antibody (68547, Cell Signaling Technology, Danvers, MA, USA; 1:1,000) were applied for protein detection as applicable. Furthermore, we used a polyclonal rabbit anti-PAX8 antibody (10336-1-AP, Proteintech, Planegg-Martinsried, Germany; dilution 1:5,000) and polyclonal rabbit anti-HNF-1β antibody (HPA002083, Atlas Antibodies, Sigma Aldrich, St. Louis, MO, USA; dilution 1:2,000). Horseradish peroxidase-conjugated anti-rabbit antibody (P0399, swine polyclonal, Agilent Technologies, Santa Clara, CA, USA; dilution 1:2,500), anti-goat antibody (P0449, rabbit polyclonal, Agilent Technologies, Santa Clara, CA, USA; dilution 1:2,500) and secondary anti-rat antibody (Jackson ImmunoResearch, Ely, UK) served as secondary antibodies. β-actin was detected by using a monoclonal anti–β-actin-peroxidase antibody (A3854, mouse monoclonal, dilution 1:60,000; Sigma-Aldrich, St. Louis, MO, USA). Signal detection was performed on an Amersham Imager 600 (GE Healthcare, Amersham, UK). Band intensity of HIF-2α and β-actin protein was quantified by densitometry using ImageJ2 software (version 2.35; National Institutes of Health). HIF-2α signal was normalized to respective β-actin loading control (please find results in Supplementary Data 12).

## Chromatin immunoprecipitation (ChIP)

Primary tubule cells, primary ccRCC and 786-0 cells were cultured in standard cell culture conditions or treated with 1 mM DMOG for 4 h as indicated. 15 cm dishes of sub-confluent cells were used and Chromatin immunoprecipitation (ChIP) experiments were performed using the ChIP-IT high sensitivity kit (Active Motif, Carlsbad, CA, USA). For immunoprecipitations, 70 µg of chromatin and 3–6 µl of antibodies directed against HIF-1α (Cay10006421, Cayman Chemicals, Ann Arbor, MI, USA) or HIF-1β (NB100-110, Novus Biologicals, Littleton, CO, USA), H3K27ac (ab4729, Abcam, Cambridge, UK) were used. Non-immunized rabbit serum or purified normal rabbit IgG (12-370, Merck Millipore, Burlington, MA, USA) served as negative controls as appropriate. Antibody–chromatin complexes were pulled down by proteinase A agarose beads (Millipore, Burlington, MA, USA). After reversal of the crosslinking by heat DNA was isolated via DNA purification columns.

## Assay for transposase accessible chromatin sequencing (ATAC-seq)

Primary renal tubule and cancer cells from three individuals as well as RCC4 and 786-0 cells were grown under standard culture conditions and harvested at 80% confluency. RCC4 cells re-expressing pVHL and U-87 cells were exposed to 1 mM DMOG for 16 h or standard culture conditions and harvested at 80% confluency. RCC4/i.VHL cells were exposed to 0.5 µg/ml doxycycline or DMSO for 18 h and harvested at 80% confluency. Cells were trypsinized, manually counted and 60,000 cells were directly subjected to the Omni-ATAC protocol as described by Corces et al.[56] Libraries were prepared using the Illumina Tagment DNA TDE1 Enzyme and Buffer Kit (20034197, Illumina, San Diego, CA, USA) and purified using the DNA Clean and Concentrator-5 Kit (Zymo Research, Freiburg, Germany) according to the manufacturer's recommendations. Following purification, library fragments were amplified by using the NEB Next High-Fidelity 2X PCR Master Mix (New

England Biolabs, Ipswich, UK) and custom made indexed/ barcoded Nextera PCR primers (Sigma Aldrich, St. Louis, MO, USA) as published by Corces et al.[56] Libraries were amplified for a total of 8–12 cycles. AMPure XP beads (Beckman Coulter GmbH, Krefeld, Germany) were used for library clean-up and double-sided size selection according to the manufacturer's protocol. DNA concentration of the libraries and fragment size distribution were determined by electrophoresis using the Agilent 2100 Bioanalyzer (Agilent, Santa Clara, CA, USA). Equimolar concentrations of barcoded libraries were pooled prior to sequencing.

## Cleavage under targets and tagmentation (CUT&Tag)

CUT&Tag was performed as described in Kaya-Okur et al.[57] with modifications based on the protocol published by Bartosovic et al.[58] Depending on the cell line, we used 250,000 to 350,000 cells per condition and antibody. Cells were washed three times in 200 µl of antibody buffer (20 mM HEPES pH 7.5, 150 mM NaCl, 2 mM EDTA, 0.5 mM spermidine, 0.05% digitonin, 0.01 % NP-40, 1× protease inhibitors and 1% BSA). Cells were centrifuged for 5 min at 600 g, resuspended in 100 µl antibody buffer with 1:50 diluted primary antibody (monoclonal rabbit anti-HIF-1β antibody [5537S, Cell Signaling Technology, Danvers, MA, USA] or polyclonal rabbit anti-HIF-1β antibody [NB100-110, Novus Biologicals, Biotechne, Wiesbaden, Germany] or monoclonal rabbit anti-H3K27ac antibody [ab177178, Abcam, Cambridge, UK] or polyclonal rabbit anti-HIF-1α antibody [Cay10006421, Cayman Chemicals, Ann Arbor, MI, USA] or polyclonal rabbit anti-HNF-1β antibody [HPA002083, Atlas Antibodies, Sigma Aldrich, St. Louis, MO, USA]) and incubated overnight. The next day cells were centrifuged for 3 min at 600 g, washed once with 200 µl of Dig-wash buffer (20 mM HEPES pH 7.5, 150 mM NaCl, 0.5 mM spermidine, 0.05% digitonin, 0.01 % NP-40, 1× protease inhibitors, 1% BSA), resuspended in 200 µl of Dig-wash buffer with 1:50 diluted secondary antibody (ABIN101961, Guinea Pig anti-Rabbit IgG antibody, Antibodies-Online, Aachen, Germany) and incubated for 1 h at room temperature with slow rotation. Afterwards cells were centrifuged for 3 min at 600 g, washed three times with 200 µl of Dig-wash buffer, resuspended in 200 µl of Dig-300 buffer (20 mM HEPES pH 7.5, 300 mM NaCl, 0.5 mM spermidine, 0.05% digitonin, 0.01 % NP-40, 1× protease inhibitors, 1% BSA) with 1:100 diluted protein A–TN5 transposase (C01070002, Diagenode, Seraing, Belgium, loaded with mosaic adapters according to manufacturer's instructions) and incubated for 1 h rotating at room temperature. Following pA-Tn5 loading cells were washed three times with Dig-300 buffer, resuspended in 200 µl of tagmentation buffer (20 mM HEPES pH 7.5, 300 mM NaCl, 0.5 mM spermidine, 0.05% digitonin, 0.01 % NP-40, 1× protease inhibitors, 10 mM MgCl$_2$) without BSA and incubated for 1 h at 37 °C. Tagmentation was stopped by addition of 7 µl of 500 mM EDTA. Subsequently 4 µl SDS 10% and 2 µl Proteinase K were added and the sample was incubated for 1 h at 55 °C. DNA isolation and clean-up were performed using the ZYMO DNA Clean & Concentrator-5 kit with a 1:5 ratio of binding buffer and 25 µl of elution buffer (Zymo Research, Freiburg, Germany). Following purification, DNA fragments were amplified by using the NEB Next High-Fidelity 2X PCR Master Mix (New England Biolabs, Ipswich, UK) and custom made indexed/ barcoded Nextera PCR primers (Sigma Aldrich, St. Louis, MO, USA) as published by Kaya-Okur et al.[57] Libraries were amplified for a total of 15 cycles. SPRIselect bead-based reagent (B23317, Beckman Coulter GmbH, Krefeld, Germany) was used for library clean-up according to the manufacturer's protocol. For H3K27ac CUT&Tag samples 5 to 40 Mio mapped reads were generated. For CUT&Tag with antibodies directed against transcription factors 1.3 to 21.4 Mio mapped reads were acquired.

## Reporter assays

668 bp and 727 bp sequences covering the hypoxia responsive elements of *EPAS1* enhancer 1 and *EPAS1* enhancer 2, respectively (hg38; chr2:46073427-46074094 and chr2:46232652-46233378) were PCR

amplified from genomic DNA of primary cells or 786-0 cells with intact or defective *EPAS1* enhancer 1 or 2 sites and ligated into the pGL3 promoter vector (Promega, Madison, WI, USA) using KPNI and NHEI restriction sites. Sequences were verified using Sanger sequencing (Eurofins Genomics, Ebersberg, Germany). Transfections of plasmids and a control β-galactosidase reporter were performed in Kelly cells using X-tremeGENE®HP DNA transfection reagent (Roche Diagnostics, Basel, Switzerland) according to the manufacturer's instructions. Transfected cells were stimulated with DMOG where applicable for 16 h before harvest. Luciferase activity in extracts was measured using the Steady-Glo luciferase reporter assay system (Promega, Madison, WI, USA). Values were normalized to β-galactosidase activity in the same cell lysates. Primer sequences are listed in Supplementary Data 11.

## Xenograft tumor model

Animal experiments were conducted under license 55.2.2-2532-2-2150-19 approved by the State of Bavaria, Regierungspräsidium Unterfranken, Germany. NOD.Cg-Prkdc[scid] Il2rg[tm1Wjl]/SzJ (NOD/SCID-gamma) mice were bred in-house in the animal facilities of the Friedrich-Alexander University of Erlangen-Nürnberg. Mice were provided with food and water ad libitum and kept at a 12:12 h light-dark cycle, 45-65 % humidity, and at 20 to 24 °C. Age-matched littermates of both sexes were used for experiments and the order of injections of cells were randomized. Cells were harvested using StemPro Accutase (Gibco, Thermo Fisher Scientific, Waltham, MA, USA) for 5 min. $5 \times 10^6$ cells were used per injection. Cells were resuspended in 50 μL PBS and 50 μL Matrigel (Basement Membrane Matrix, 354234, Corning, Bedford, MA, USA) and kept on ice. We used a 30 G insulin syringe (BD, Franklin Lakes, NJ, USA) for subcutaneous injection in the right flank of 8–12-week-old NOD/SCID-gamma mice in a randomized, non-blinded manner. The tumor growth was monitored twice a week using caliper measurements by an investigator blinded for the different conditions. Tumor volume was calculated using this formula: volume = (length × width2)/2. Mice were euthanised by cervical dislocation and tumors dissected at week 10 of the observation period. No tumor reached a volume of 1 cm$^3$, which was prespecified as the maximal burden.

## RNA-seq analysis

RNA was isolated using the peqGold total RNA kit (VWR Peqlab, Erlangen, Germany). Adequate quality of isolated RNA was confirmed using the RNA Integrity Number (RIN, Supplementary Data 13) determined by a Bioanalyzer 2100 (Agilent, Santa Clara, CA, USA). Samples were sequenced in technical duplicates by Novogene (Cambridge, UK) on an Illumina Novaseq 6000 or Illumina NovaSeq X plus platform to a 2 × 150 pair-end format with an average number of 20–30 million reads per sample (Supplementary Data 13). After quality control using FastQC (v0.11.8) (https://www.bioinformatics.babraham.ac.uk/projects/fastqc/) reads were aligned to the human reference genome (hg38) using the STAR alignment software (v2.7.10a)[59]. A count table of mapped reads was generated with the feature-counts software (v1.6.1)[60]. Transcripts with less than 10 reads were discarded. The DESeq2 package version 1.32.0 was used for logarithmic transformation of the data and for data exploration[61]. Differential expression analysis was performed by using the DESeq2 lfc Shrink approach. Adjusted *p*-values were calculated using the Benjamini–Hochberg method within DESeq2. Gene annotations were added to the result files using biomaRT (v2.48.3)[62]. Differentially expressed (DEG) genes were visualized with a Volcano plot.

## ATAC-seq analysis

Barcoded amplicons were sequenced on a HiSeq 2500 (Illumina) or Illumina NovaSeq X plus platform to a 2 × 150 pair-ended format (Novogene, Cambridge, UK). Reads were quality filtered according to the standard Illumina pipeline and de-multiplexed. Fastq file were processed by trimming Illumina adapter sequences using Trim Galore (v0.6.6). Reads were aligned against human reference genome (hg38) using Bowtie2 (v2.3.4.1)[63]. Using SAMtools (v1.8) aligned reads with a Mapping Quality (MAPQ) below 30 and those mapped to ChrM were removed[64]. PCR-duplicates reads were excluded via Picard (http://broadinstitute.github.io/picard/) (v2.25.2). ENCODE hg38 blacklisted regions (access 20.10.2020) and chromosome Y were removed using GenomicRanges (v1.46.1) and filterChromosomes function from regioneR (v1.26.1)[65,66]. Quality metrics are listed in Supplementary Data 14.

For analysis of differentially open regions (DOR) in ccRCC tumor cells versus primary tubule cells sequencing duplicates of each ATAC sample of three different patients were examined. For the detection of DOR between HNF-1β knock-out cells or PAX8 knock-out cells *versus* cells treated with non-targeting controls, sequencing duplicates of two single clones of cells were used per condition. Peaks were called using MACS2 (v2.2.7.1) with the following parameters "--shift -75 --extsize 150 --nomodel --call-summits --nolambda --keep-dup all -p 0.01"[67]. Reads in the peaks were normalized using the variance-stabilized transform (VST) function and batch effects were removed using the limma::removeBatchEffect function[68]. Two meta-files covering open regions in PTC or ccRCC cells, respectively, were generated using the summit approach as published by Corces et al.[37]. Differentially open regions were discovered using the DESeq2 lfcShrink approach (DESeq2 package v1.28.1)[61]. Adjusted *p*-values were calculated using the Benjamini–Hochberg (BH) method. ATAC sites exhibiting a change of accessibility with a log$_2$ fold change > 0.5 or < -0.5 with an adjusted $p < 0.05$ were defined as DOR.

SAMtools allowed merging of technical and biological replicates[64]. Normalized bigWig tracks for visualization were generated using BamCoverage (v3.3.2)[69].

Single bigWig tracks per tumor of ATAC-seq data from TCGA tumors were downloaded from the TCGA resource website https://gdc.cancer.gov/about-data/publications/ATACseq-AWG (access 28.05.2021), converted to bedGraph files, normalized using Wiggletools 1.2.11 and merged[37,70]. The bedGraphToBigWig tool was applied to acquire an integrated KIRC bigWig track for visualization[71].

## ChIP-seq analysis

Library preparation of immunoprecipitated DNA and ChIP-sequencing was performed by Novogene (Cambridge, UK). Amplicons from the ChIP-Library preparation were sequenced on an Illumina Novaseq 6000 platform to a 2 × 150 pair-ended format. Raw sequencing files in fastq format were generated via fastq-dump (v2.8.0) and adapter sequences were trimmed using Trim Galore (v0.6.6). Quality control was conducted with FastQC (v0.11.8). Subsequently, reads were mapped to the hg38 version of the human genome using Burrows-Wheeler Aligner (BWA-mem; v0.7.17-r1188)[72]. SAMtools (v1.8) removed aligned reads with a Mapping Quality (MAPQ) below 30 and those mapped to ChrM[64]. PCR-duplicates reads were excluded via Picard (http://broadinstitute.github.io/picard/) (v2.25.2). Technical replicates from the samples were merged using SAMtools. We excluded ENCODE blacklisted regions from merged files and generated normalized BigWigs for visualization using BamCoverage (v3.3.2)[65,71]. Peaks were identified applying MACS2 (v2.2.7.1) using the parameters " f BAMPE -n Input file -q 0.05" for detection of transcription factor binding sites and "f BAMPE -q 0.01 -B -n Input file -q 0.05" for analysis of histone marks[67]. Quality metrics are listed in Supplementary Data 15.

In order to generate a Meta-HIF ChIP-seq data set, published raw sequencing data of HIF-1α, HIF-1β and HIF-2α ChIP experiments in RCC4, 786-0 and primary ccRCC cells were downloaded (see Data availability) and processed as described above[9,32,73]. Peaks were combined using the GenomicRanges' "reduce" function[66]. Only HIF binding sites present in at least 4 out of 7 ChIP-seq experiments were kept for further analysis (Supplementary Data 2).

H3K27ac ChIP-seq data for differentially active enhancer and promoter regions in ccRCC tumors in comparison to normal kidney

tissue were from Yao et al.[32]. Liftover of coordinates of altered enhancers and promoters was performed by the LiftOver tool within the UCSC Genome Browser to acquire an hg38 compatible bed file for visualization[74]. After filtering for chromosome 2, average $\log_2$ fold chance and $-\log_{10}$ adjusted $p$-values were visualized with a Volcano plot using the R package EnhancedVolcano (v1.22.0). To be considered overlapping, differentially active sites must overlap with KIRC-specific ATAC sites at at least one base.

## Transcription factor motif enrichment analysis

DOR exhibiting an increased accessibility in ccRCC cells in comparison to PTC ($\log_2$ fold change > 0.5, adjusted $p < 0.05$) were filtered for overlapping H3K27ac ChIP-seq signals. These sites underwent motif analysis using HOMER2 (v4.9.1)[75]. RNA-seq data of PTC and ccRCC cells were analysed for differentially expressed genes applying the DESeq2 lfcShrink approach (DESeq2 package v1.28.1)[61]. Integration of motif discovery in DOR and differential expression of respective transcription factors allowed for calculation of the Motif enrichment score according to Xin et al.[76]. To determine transcription factor motifs in DOR with decreased accessibility ($\log_2$ fold change < -0.5, adjusted $p < 0.05$) in single clones of 786-0 HNF1-ß knockout versus single clones of control (nt) cells and single clones of 786-0 PAX8 knockout versus single clones of control (nt) cells HOMER2 (v 4.9.1) was applied. Results were plotted using ggpubr (v.0.4.0).

## ABC-analysis

The Activity by Contact (ABC) Model of Enhancer-Gene Specifity[30] (v1.0) was applied to predict enhancers that regulate *EPAS1* expression in primary tubule and ccRCC cells. Inputs for the individual-specific analysis were bam files for ATAC-seq and H3K27ac ChIP-seq as well as TPM-normalized RNA-seq data from corresponding PTC and ccRCC cells of 3 different patients. A similar analysis was performed on published data from 786-0 cells[31–33]. For quantification of the contact frequency publicly available average Hi-C profiles (https://github.com/charlesfulco/ABC-Enhancer-Gene-Prediction)[30] were used as previously described by others[77]. Only enhancer predictions with Power-law scores ≥ 0.015 were kept.

## Correlation analysis ATAC vs RNA-seq

Cancer type-specific normalized ATAC-seq counts matrices from individual KIRC and LGG samples were downloaded from https://gdc.cancer.gov/about-data/publications/ATACseq-AWG[37] (access 14.06.2021). Normalized RNA-seq data (TPM) for KIRC and LGG were downloaded from the TCGA database using TCGAbiolinks (v.2.22.4)[78] (access 13.06.2023). A Spearman's rank correlation analysis for the indicated ATAC sites was performed and plotted using ggprubr (v0.4.0).

## Quantification of the H3K27ac ChIP-seq signal in predefined genomic regions

ChIP-seq bigWig (bw) files were used to quantify the H3K27ac signal at predefined genomic regions. These regions included the *EPAS1*-enhancer 2 locus (original coordinates in hg19, liftover to hg38: chr2:46222961-46234261) and a HIF-binding *NDRG1* enhancer (original coordinates in hg19, liftover to hg38: chr8:133337907-133395407) as predefined by Yao et al.[32]. BigWig signals for the respective regions were extracted using the import function of the rtracklayer package (v1.54.0). After extraction, an approximation of the integral using the Riemann sums method was calculated[79]. The resulting integral values (reads per genomic content, RPGC) were then used to compare the H3K27ac signal between control and tumor tissues. Additionally, the integral values from both control and tumor tissues were correlated with the respective HIF-2α mRNA expression (TPM) employing the Spearman's rank method using the stat_cor function. Results were visualized with ggpubr (v0.4.0).

## Gene set enrichment analysis

Gene set enrichment analysis (GSEA) used pre-ranked genes and applied 100,000 permutations. Pre-ranking of differentially expressed genes was based on the fold-difference as determined by DESeq2[61]. Functional enrichment analysis was performed via fgsea (v1.20.0)[80]. The "Hallmark" gene sets were downloaded from MsigDB (v7.5.1)[81]. Adjusted $p$-values were calculated using the Benjamini–Hochberg method. Pathways with an adjusted p-value < 0.05 were chosen for visualization using ggplot2 (v3.3.6) or fgsea.

## CUT&Tag-seq analysis

CUT&Tag libraries were sequenced by Novogene (Cambridge, UK) on an Illumina NovaSeq XPlus platform to a 2 × 150 pair-ended format. FastQC (v0.11.8) was applied for quality control and sequencing adapters were removed using Trim Galore (v0.3.3). Bowtie2 (v2.3.4.1) was used for read alignment against the human reference genome (hg38). Low quality reads (MAPQ < 30) and reads mapping to mitochondrial DNA or ENCODE blacklisted regions were eliminated using SAMtools (v.1.8)[82]. Picard (http://broadinstitute.github.io/picard) (v.2.25.2) removed PCR-duplicate reads. BamCoverage (3.3.2) was used to create normalized bigwig tracks for visualization. Peaks were called via MACS2 (v2.2.7.1) using the MACS2 callpeak function. We benchmarked the results of the HIF-1β CUT&Tag experiment against published HIF-1β ChIP-seq peaks in the same cell line[45]. ChIP-seq data for HIF-1β (GEO Accession: GSE34871) were downloaded from https://chip-atlas.org/ (SRX114494, q < 1E-10). Overlap analysis of the 805 ChIP-seq HIF-1β peaks resulted in 543 of these sites with significant CUT&Tag signals. Top ranked CUT&Tag peaks showed a remarkable overlap with the ChIP-seq validated HIF-1β binding sites (Supplementary Fig. 26).

For comparison of H3K27ac levels under different conditions, samples from RCC4/i.VHL-HA cells were sequenced in technical duplicates and tracks were merged for visualization. Captured DNA from two single control (nt) or HIF-1β knock-out clones of cells from each cell line (RCC4 or 786-0 cells) were sequenced in technical duplicates. All tracks belonging to one condition were merged for visualization.

For the comparison of HIF-1β and H3K27ac peak signals in single clones of 786-0 cells with a defective or intact E2_HIF enhancer KIRC_10770 element, two metafiles (HIF-binding H3K27ac sites and Non-HIF binding H3K27ac files) were generated. Peak calling on H3K27ac CUT&Tag tracks of the aforementioned single clones of cells was performed using the MACS2 tool. DiffBind (v. 3.4.11) was applied to analyze the H3K27ac peaks and to establish a consensus peak set (n = 33,027)[83]. These consensus peaks were then aligned to pre-defined and published HIF-1β binding sites (GEO Accession: GSE34871, downloaded from ChIP-Atlas: SRX114494, q < 1E-10)[45] to identify regions overlapping HIF-1β binding. To quantify the HIF-1β or H3K27ac CUT&Tag signal at the center of the H3K27ac consensus peaks with a HIF-binding site ($n = 472$) or without a HIF-binding site (non-HIF binding, H3K27ac consensus peaks, $n = 32,555$), bigwig files for HIF-1β and H3K27ac CUT&Tag signals from biological replicates of each condition (nt vs E2_HIF ko) were combined. Peak signal intensity was analyzed using the computeMatrix function and results were visualized with the plotProfile function from the deepTools suite (v. 3.5.1).

## Expression quantitative trait loci (eQTL) analysis

TCGA level 3 RNA-seq data for KIRC, KIRP or KICH tumors was coupled with level 2 genotyping data of the respective patients from the Affymetrix Genome-Wide Human SNP Array 6.0. SNP reference rs11894252 was mapped to SNP_A-2290064 and rs12617313 to SNP_A-4200988 in the Affymetrix array. Data from patients with low-confidence SNP calls (confidence <0.01) were excluded. Hence, 387 patients were retained for rs11894252 and 403 patients for rs12617313 in the KIRC cohort. In the KIRP cohort 225 patients for rs12617313 were included. The KICH

cohort comprised 23 patients for rs12617313. In order to evaluate the significance of association between the SNP genotype and expression of *EPAS1*, RNA-seq expression across the samples was fitted to a negative binomial generalized linear model against the genotype status. Hereafter, the likelihood ratio of this model versus a model that ignores genotype status was computed. Finally, a $\chi^2$-test was used to identify the significance of the genotype coefficients in stratifying the patients[43].

## Co-accessibility analysis using Cicero

Single nucleus ATAC-sequencing data and metadata were downloaded from GEO (GSE240822)[35]. For each snATAC tumor sample, cells, which were not listed in the provided metadata, were removed. Peaks were called using MACS2 (version 2.2.7.1) by applying the function „CallPeaks" of the R package Signac (version 1.12.0)[84]. Peaks overlapping genomic blacklist regions were excluded using Signac's list „blacklist_hg38_unified" and the function „subsetByOverlaps" (invert = TRUE). We obtained a common peak set by merging intersecting peaks across samples („reduce" from R package „GenomicRanges", version 1.50.2) and removed resulting peaks that had a width greater than 10,000 or smaller than 20 bp. Finally, datasets were transformed into peak-count matrices using the function „FeatureMatrix". Datasets were merged into one object for Signac downstream analysis, i.e., term frequency-inverse document frequency (TF-IDF) normalization („RunTFIDF") and selection of most frequently observed features („FindTopFeatures") for dimension reduction via singular value decomposition („RunSVD"). We used latent semantic indexing (LSI) components 2:30 to compute the Uniform Manifold Approximation and Projection (UMAP) with function "RunUMAP" of R package „Seurat" (version 5.0.1)[85]. The dataset was further subset to cells that were annotated as cell type „Tumor" by the provided metadata and co-accessibility analysis restricted on chromosome 2 was performed using function „run_cicero" of the R Package Cicero (version 1.3.9)[40].

## CRISPRi screening data

Source data from Patel et al.[25] were downloaded from https://zenodo.org/record/6335339 (access 06.02.2023). The provided R script was used to plot the individual enhancer sensitivity score for each construct. Shown are empirical one-sided p-values calculated for each target region by 10000 permutations of the normalized sgRNA depletion scores for each tumor as published by Patel et al.[25].

## Liftover of capture-C data

Liftover of published Capture-C data and ChIP-seq data from hg19 to hg38 was performed using CrossMap (v0.6.1)[86].

## Statistical analysis

All statistical analyses excluding sequencing data were performed using GraphPadPrism Version 9.0.2 (GraphPad Software Inc., San Diego, CA, USA). Results are presented as mean +/- SD if not indicated otherwise. Statistical tests applied to determine significance are provided in the figure legends. The two-tailed unpaired Student's t-test was used for comparison of two conditions. One- or two-way ANOVA analysis followed by pairwise or multiple comparisons were performed for experimental setups comprising more than two conditions or two influencing factors. Adjusted *p*-values were determined using the Bonferroni correction unless otherwise indicated. The one-sample t-test comparing the mean with a hypothetical value of 1 was used if the values of the control condition were set to 1. Statistical testing of NGS datasets are described in the respective methods section.

## Visualization

BigWig files were visualized using the Integrative Genomics Viewer (IGV_2.8.4). Adobe Photoshop Elements (Adobe Inc., San Jose, CA, USA) was used for creating figures. Schematics were created using Biorender.com: Naas, S. (2025) https://BioRender.com/c87s173 (Fig. 1e) and Naas, S. (2025) https://BioRender.com/w24j560 (Figs. 5f, 6a, 7i).

## Reporting summary

Further information on research design is available in the Nature Portfolio Reporting Summary linked to this article.

## Data availability

ATAC-seq, ChIP-seq, CUT&Tag-seq, and RNA-seq data generated for this project have been deposited in the GEO database under accession number GSE256001. Access to unprocessed raw data of primary cells is restricted to protect the privacy and intent of research participants consistent with the written informed consent agreements provided by individual research participants. We used the following publicly available data sets (GEO Database): Capture-C data in 786-0: GSE130988[34], https://www.ncbi.nlm.nih.gov/geo/query/acc.cgi?acc=GSE130988, HIF-1β and HIF-2α ChIP-seq data in 786-0: GSE67237[9], https://www.ncbi.nlm.nih.gov/geo/query/acc.cgi?acc=GSE67237, HIF-1β ChIP-seq data from 786-0: GSE34871[45], https://www.ncbi.nlm.nih.gov/geo/query/acc.cgi?acc=GSE34871, HIF-ChIP-seq in ccRCC cells and H3K27ac ChIP-seq in normal kidney and tumor tissue as well as 786-0 cells: GSE86095[32], https://www.ncbi.nlm.nih.gov/geo/query/acc.cgi?acc=GSE86095, ATAC-seq in 786-0 cells: GSE102807, https://www.ncbi.nlm.nih.gov/geo/query/acc.cgi?acc=GSE102807, HIF-1β ChIP in T47D, A549, HCT116 and PC3 cells: GSE130989[34], https://www.ncbi.nlm.nih.gov/geo/query/acc.cgi?acc=GSE130989, HIF-1β ChIP in HepG2, HKC-8 and RCC4: GSE120885[73], https://www.ncbi.nlm.nih.gov/geo/query/acc.cgi?acc=GSE120885, HIF-1β ChIP in HUVEC: GSE89836[87], https://www.ncbi.nlm.nih.gov/geo/query/acc.cgi?acc=GSE89836, HIF-1β ChIP in Hela: GSE159128, https://www.ncbi.nlm.nih.gov/geo/query/acc.cgi?acc=GSE159128, HIF-1β ChIP in PTC #4: GSE101063. https://www.ncbi.nlm.nih.gov/geo/query/acc.cgi?acc=GSE101063, RNA-seq in 786-0: GSE115389[31], https://www.ncbi.nlm.nih.gov/geo/query/acc.cgi?acc=GSE115389, RNA-seq in 786-0-xenografts with or without PT2385 treatment: GSE153711[46], https://www.ncbi.nlm.nih.gov/geo/query/acc.cgi?acc=GSE153711, snATAC-seq data and metadata: GSE240822[35], https://www.ncbi.nlm.nih.gov/geo/query/acc.cgi?acc=GSE240822. Sequencing data was re-analyzed and mapped to hg38 as outlined above. KIRC regulatory elements from TCGA were downloaded from https://gdc.cancer.gov/about-data/publications/ATACseq-AWG (access 24.03.2021). RSEM-normalized RNA-seq data from TCGA were downloaded using RTCGA.rnaseq (v.0.1-3) (access 28.05.2021). Data for significantly hypomethylated CpGs in KIRC were accessed via http://www.bioinfo-zs.com/smartapp/ (28.11.23)[36]. TCGA data for the methylation status of individual CpGs at the *PRKCE-EPAS1* locus in the KIRC dataset (normal and tumor tissue) were downloaded from http://maplab.imppc.org/wanderer/ (access 12.04.2024)[88]. Copy number alterations and corresponding RNA-seq data from the TCGA PanCancer Atlas were downloaded via the cBio-Portal (https://www.cbioportal.org/, original data: https://gdc.cancer.gov/about-data/publications/pancanatlas, access 20.07.2023)[89]. SNPs with cis-eQTLs for KIRC were extracted from https://gonglab.hzau.edu.cn/PancanQTL[44]. Data from 733 biosamples were published by Meuleman et al. and were accessed via a public server (https://index.altius.org/, access 03.08.2023)[41]. Sequencing data and analysis form CRISPRi screen for oncogenic-HIF-2-bound enhancer sites in 786-M1A cells[25] were downloaded from https://zenodo.org/records/6335339 (access 06.02.2023). Transcription factor motifs were downloaded from the JASPAR 2024 database (access 03.12.2024). Access to the restricted TCGA genotyping data for the KIRC, KICH and KIRP cohorts presented in the current publication was granted via phs000178 at the dbGaP web site. These data were generated by the TCGA Research Network and can be accessed via https://www.cancer.gov/tcga (access KIRC data: 31.10.2024, access KICH and KIRP data: 11.12.2024). The

remaining data are available within the Article, Supplementary Information or Source Data file. Source data are provided with this paper.

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

## Acknowledgements

We thank Stephanie Palffy, Margot Rehm, Astrid Ebenau-Eggers, Pia Klanke, Barbara Teschemacher and Johanna Stöckert for excellent technical assistance. We thank the Comprehensive Cancer Center Erlangen-EMN at the Uniklinikum Erlangen and PD Dr. Markus Eckstein for his support in the histopathological evaluation of kidney tumors. Parts of this work were performed by René Krüger in fulfillment of the requirements for obtaining the degree Dr. rer. hum. biol. S.N. was supported by the Interdisciplinary Center for Clinical Research (IZKF) at the University Hospital of the University of Erlangen-Nuremberg (Junior Project J114). The study was funded by the German Research Foundation (Projektnummer 509149993 TRR374, Projektnummer 467519218, and SCHO 1598/2-1 to J.S.). J.N. is a member of the Vienna Biocenter PhD Program, a Doctoral School of the University of Vienna and Medical University of Vienna, Austria, and is supported by the Austrian Science

Fund (Special Research Program SFB-F78, F 7811-B) and the Research Platform SinCeReSt- Single cell regulation of stem cells. We thank the Verein VHL (von Hippel-Lindau) betroffener Familien e.V. for financial support of this study.

## Author contributions

Stephanie Naas: investigation, data curation, writing—original draft; René Krüger: software, data curation; Steffen Grampp: investigation; Victoria Lauer: investigation; Andre Kraus: data curation; Julia Naas: software, data curation; Fabian Müller: resources; Franziska Gsottberger: resources; Mario Schiffer: resources; Bernd Wullich: resources; Arndt Hartmann: resources; Marc Stemmler: investigation; Johannes Schödel: conceptualization, writing—original draft, supervision, funding acquisition, data curation. All authors approved the final version of this manuscript.

## Funding

## Competing interests

The authors declare the following competing interests: J.S. has received honoraria from MSD Sharp & Dohme GmbH. All other authors declare no competing interests.
