## [Transparent Peer Review file · Nature Communications]

HIF sustain a transcriptional regulatory circuit of EPAS1 expression in renal clear cell carcinoma

Corresponding Author: Professor Johannes Schödel

Version 0:

Reviewer comments:

Reviewer #1

(Remarks to the Author)

Summary:

SNPs in the intronic region of the gene EPAS1 (encoding for HIF2 α), has been positively associated with the risk for renal cell carcinoma, yet how HIF2 α is regulated to sustain ccRCC is poorly understood. In this manuscript Naas et al have used epigenomic profiling to identify an enhancer that controls the expression of EPAS1 specifically in ccRCC. The authors show that this enhancer is crucial to the expression of EPAS1, and they suggest that the activity of this enhancer is regulated by HIF and other renal-specific transcription factors. Through this study, the authors have provided a molecular mechanism explaining how the oncogene, HIF2 α , is controlled in ccRCC to contribute to accelerated tumor formation/growth.

Major Concerns:

- Figure 1: The authors have shown a representative RNAscope image for EPAS1 in ccRCC tissue (1e). The authors should include images from the other three patients in supplementary figures as well. To also ensure that the signal is specific, the authors should consider including a negative control (either normal tissue or control PTC cells). If the RNAscope probes have been validated in a different study, the authors should include the reference. Could the authors also elaborate on how cancer cells were distinguished from noncancer cells in 1e? Why is the EPAS1 signal restricted only to some areas within the tissue and is this seen across patients?

- Figure 2: The authors have highlighted enh-1031503 as ccRCC-specific. However, this region is also active in PTC from patients 1 and 2, less so in patient 3. Could the authors comment on why this may be? Could it be cross-contamination or some other reason? If the authors perform differential analysis on the K27ac peaks at this region, is there significant difference? The same variability in H3K27ac signal is apparent in Supp Fig 3, when comparing normal and ccRCC tissue. Could the authors elaborate on this variability? Perhaps include the corresponding EPAS1 expression status?

In fig 2e, the authors should also include H3K27ac signal, wherever possible, to compare the degree of enhancer activity across these tumor types.

- Figure 3: When the authors knock-out HIF1 β (3f, g), there is a modest decrease on HIF2 α . Is this modest decrease in line with what is expected? Is this true even at the EPAS1 transcript level? The authors have shown how ATAC signal changes at the ccRCC-enhancer after re-introducing VHL (Fig 3e). The authors should also depict how the enhancer activity is changing as VHL is being re-expressed. Additionally, the authors show how EPAS1 RNA levels change upon addition of DMOG. Can the authors include how the EPAS1 enhancer status is changing in this case?

- Figure 4: It is unclear if the HIF ChIP-seq signal in 4a is from WT/untreated PTC cells or cells treated with DMOG. Can the authors point out where exactly KIRC_10770 and KIRC_10767-69 lie? Are the HIF peaks specifically on KIRC_10770? Is there some difference in motifs or methylation status at enhancer 1 versus enhancer 2 that could explain why enhancer 1 shows stable HIF binding while enhancer 2 shows more dynamic binding? Based on Fig 4d, it appears as though PAX8 and HNF1 peaks do not coincide on KIRC_10770. What is the level of change in H3K27ac levels when these TFs are knocked-out in 786-O cells? Can the authors include this data to strengthen their claim that PAX8 and HNF1 β regulate HIF2 α , and that enhancer 1 or 2 maybe involved?

- Figure 5: The authors have deleted E1 or/and E2 to assay for the downstream effect on EPAS1 expression. However, there is insufficient data provided to show that the enhancers have been effectively abrogated. Can the authors include genotyping and DNA sequencing data to show that the HIF motifs are deleted? Can they also include H3K27ac data to show the extent to which the enhancer activity has been affected upon CRISPR-manipulation? It is curious that even when E1+E2 are deleted, HIF2 α is still expressed. Could the authors speculate on why that might be? It is unclear to me, what the authors mean by enhancer 2 activity in trans. Do the authors mean to assess the effect of deleting enhancer 2 on HIF-2 α downstream gene targets? Or did the authors only delete the enhancers on one of the alleles? Are 786-O cells diploid for the EPAS1 locus? If no, how many alleles for E1 and E2 were successfully deleted by the authors? Could the authors include these details? Although it is mentioned in the methods, in the figure and figure legend, the authors state that the CUT&TAG was performed to assess HIF1 β binding.

- Based on the ATAC-seq data, the element KIRC_10770 appears to be the most dynamic. What is the direct contribution of this element in regulating EPAS1 expression levels? In the CRISPR deletions performed in Fig 5, it is unclear if this element was abrogated or not. The authors should consider specifically deleting KIRC_10770 to assay for downstream effects. Also, does this element carry motifs for HIF? Or for Pax8/HNF1 β ? Surprisingly, contrary to the data from this manuscript, analysis in Supp 6b shows that KIRC_10770 is very modest DNase hypersensitivity in the renal/cancer category. Is it possible for the authors regroup this data to separate out cancer v/s non-cancer samples?

Minor Concerns:

- The authors can include quantification for the western blots to depict the degree of change in HIF2 α levels upon the different perturbations throughout the manuscript.
- Fig 1d is not necessarily adding to the figure and can be removed or moved to supplementary.

Reviewer #2

(Remarks to the Author)

HIF-2a is one of the most important driving factors in clear cell renal cell carcinoma.

Naas et al. present a very interesting manuscript that identifies a feed-forward regulatory circuit controlling HIF-2a (EPAS1) mRNA expression in ccRCC. This involves an enhancer element that regulates HIF-2a mRNA expression.

The study makes excellent use of human cell resources from normal kidney tissue and from ccRCCs, of cultured ccRCC and other cancer cell lines, conducts different analyses of chromatin structure, status and protein binding (ATAC-seq, ChIP-seq, CUT&TAG), combined with RNA expression analyses and functional mutation of identified genomic sites as well as of the transcription factors in the regulatory network. The study also integrates many different publicly available datasets from previous studies to support the experimental work (however, many of the figures are in fact entirely or heavily based on already published data). While the data presented in the manuscript suggest that the overall mechanism that the authors have uncovered is very likely to be true, there are several aspects of the study that remain incomplete to fully support the proposed model and conclusions, as well as to address some important open questions that are central to the overall findings. Most of these could be readily addressed using the tools that the authors describe in the manuscript.

Major issues

1. A very interesting observation of the overall study is that the HIF-2a enhancer E2 appears to be relatively specific to ccRCC (and also to glioblastoma) which might explain in part the sensitivity of the kidney to the oncogenic effects of HIF-2a. However, there are several limitations to the current study that prevent this conclusion being made. Fig 3b-e and Suppl Fig 8 nicely show that VHL re-introduction into ccRCC cells suppresses HIF protein abundance, HIF-2a mRNA abundance and loss of ATAC-seq signal at the enhancer, showing that there is a dependency on VHL in established cancer cell lines. However, efforts to show the dependency of this activity on HIF fall short in their current form. Experiments in Fig 3f and g would need to be complemented by HIF-2a mRNA measurements and crucially by ATAC-seq to investigate the status of the enhancer. Enhancer status assessment is also needed for experiments in Fig 4f and g which investigate the effects of mutation of PAX8 and HNF-1b. It remains unclear whether PAX8, HNF-1b or HIF factor binding is mutually dependent and what effect these factors have on the accessibility of the enhancer.

2. The overall question which needs to be addressed is how the enhancer becomes accessible in ccRCC. It goes from a state of inaccessibility in normal renal proximal tubule epithelial cells to being accessible in ccRCC cells. Based on the presented evidence it appears likely that this transition might be (at least partly) induced by loss of VHL and be HIF-dependent, however other genetic/epigenetic alterations that have arisen in these cancer cell lines might have also cooperated with VHL loss to induce this state (given the very long timeframe of ccRCC evolution this is plausible). This concept should be investigated. Is VHL loss and HIF-a activation necessary and/or sufficient for enhancer activation? For example, it should be investigated if mutation of VHL or overexpression of non-degradable HIF-1a and HIF-2a mutants in normal renal epithelial cells alters endogenous HIF-2a mRNA levels and enhancer status? It would also be highly interesting to conduct hypoxia time course experiments in normal renal cells as hypoxia has previously been shown to induce a transient HIF-1a increase, followed by a longer-term HIF-2a increase. It may be that the feed-forward enhancer mechanism identified here plays a role in this effect.

3. Fig 3h and 3i lack evidence that DMOG was active in these cells (e.g. HIF-a western blots or activation of well characterised HIF-target genes) and proof of functional knockdowns is lacking.

4. Fig. 3b, 3c. The argument presented around these figures is not convincing as it relies on the absence of an effect to prove a hypothesis. There is such a massive spread of expression levels in TCGA data across each of the three genotypes that it appears very unconvincing that there is any relationship between these genotypes and HIF-2 α mRNA.

5. Fig 5. The effects of the sgRNAs at the target loci in whole populations and single cell clones should be verified by DNA sequencing.

6. Supplementary Figure 6d is difficult to understand and currently not possible to interpret. A diagram of the vectors that are being tested would be helpful. It is not clear what normalized luciferase intensity is referring to as there is no mention of the empty luciferase vector. From this it appears that the normalization is simply a transfection normalization (beta-galactosidase) but luciferase signals are not relative to an empty vector. It is therefore impossible to conclude whether DMOG treatment has any specific effects on the E1 and E2 elements or whether this is a non-specific effect on the luciferase plasmid in general. Cloning HRE mutants into the E1 and E2 elements would provide important additional evidence that the E1 and E2 elements are responsive to HIFs.

7. Overall the study lacks in vivo evidence. Figure 6 essentially represents a series of reanalyses of published data (b,c,d) that correlate with the model that is being proposed by the current study. 6e and f show a cell culture 3D growth assay presumably as a surrogate for a xenograft assay comparing control clones and E2 ko clones but there is no evidence that the observed growth differences are dependent on the enhancer or on changed HIF-2 α expression levels. A rescue experiment with exogenous HIF-2 α expression would be beneficial to support the dependency of this assay on HIF-2 α levels. It is currently unclear what the experiments shown in Fig 6 add compared to the published studies and don't even directly validate the enhancer hits that were identified in the published screen from the Nature paper (reproduced in Fig 6d).

Minor issues

1. Some of the diagrams that are presented as figure panels are not necessary to understand these simple experiments. Figs 1d, 1g, 2a, 5f could be removed from the figures.
2. While many figures use data mining to interrogate datasets generated in the current study, quite a few figure panels are simply re-representations of data generated in other studies so should not be shown again. e.g. the representation of TCGA data shown in Fig 1b has been previously published by Nature Communications in another study (<https://doi.org/10.1038/s41467-020-17873-3>). Fig 6b and 6c are re-representations of published figures in Nature ([doi:10.1038/nature19796](https://doi.org/10.1038/nature19796)) and Fig 6d replots data obtained in a recent Nature publication (<https://doi.org/10.1038/s41586-022-04809-8>).
3. The axes of Fig 2d are not labelled in the figure.
4. The western blot lanes in Fig 3c should be labelled above the blots as it is not immediately clear that these are the same conditions as in 3d below.
5. It would be helpful for the DNA nucleotide HRE sequences that are being targeted in Fig 5a to be shown in relation to the sgRNAs that target these sequences so that the reader can understand what type of mutations are being introduced
6. Supplementary Fig 6. The authors state that a small fraction of non-ccRCC cell lines show signal at enhancer 2 as further evidence of the ccRCC specificity of enhancer binding of HIF-b. However, 3 of 8 cell lines show this signal so this statement is not supported by evidence

Reviewer #3

(Remarks to the Author)

HIF has been known to be important in the development and progression of renal clear cell carcinoma though its mechanism of dysregulation in the cancer is largely unknown. In the article, Naas et al. used epigenome and transcriptome data to study the mechanism of augmented EPAS1 expression in ccRCC. They found that HIF-2 α is highly expressed in ccRCC compared to normal samples. According to the enhancer-gene linkage inference, the ccRCC-specific enhancer enh-1031503 shows great potential to regulate gene EPAS1 expression through the binding of HIF, HNF, and PAX8 transcription factors. They have also verified that knocking out the enhancer can inhibit the development of ccRCC, which indicates a great potential for clinical application for the study. I also have a few comments:

Major comments

1. I understand this is a new mechanism of HIF-2 α regulation, could you also explain the impact and potential application of the study? What is the advantage of blocking E2 instead of eliminating HIF-2 α expression?
2. In vivo and survival studies shall be considered to prove the impact of E2 regarding its impact on cancer development and progression.
3. It seems most conclusions are based on 786-0 cell lines (e.g. ChIP-seq data, KO experiment, and xenograft model), which may be biased. Could you also provide the result of RCC4 to make the conclusion more solid? Also, a ccRCC cell line without VHL loss can be a control.

Minor comments

1. Is EPAS1 expression related to the survival of patients?
2. When you have single-cell ATAC-seq data for ccRCC, could you also verify if E2 is co-accessible with other regulatory

elements such as E1 and promoter of EPAS1?

3. It also requires data from corresponding patient samples such as Figure 4a.

4. It seems HIF, HNF, and PAX8 bind to different regulatory elements in 4d. Do you have any direct experiment evidence for the co-binding / collaboration at E2? What if designing sgRNA to target PAX8 or HNF binding sites at E2?

5. 5b lacks the impact of individual E1 and E2.

6. Is the SNP rs10193766 a risk site of ccRCC?

Reviewer #4

(Remarks to the Author)

Version 2:

Reviewer comments:

Reviewer #2

(Remarks to the Author)

Many thanks to the authors for conducting so many new experiments that were suggested by myself and the other reviewers. The quality of the data and presentation of all of the figures is very high. I believe that these fully address all of the questions that were raised during the review and that the new submission represents a complete story. The study identifies the transcriptional regulatory mechanisms that underlie the tissue-specificity and oncogenicity of HIF-2 α expression in ccRCC. I congratulate the authors on their work and believe that this represents a very important contribution to understanding the mechanisms that underlie the development and maintenance of ccRCC and provides an explanation for the tissue specificity of the consequences of VHL mutation. This work makes an extremely important contribution to the research field of ccRCC.

Reviewer #3

(Remarks to the Author)

All my comments have been addressed.

Reviewer #4

(Remarks to the Author)

We thank the referees for both their positive impressions of our work, and their helpful comments which improved the manuscript and data presentation. Please find our responses in blue.

REVIEWER COMMENTS

Reviewer #1, expertise in CUT&Tag and chromatin biology (Remarks to the Author):

Summary:

SNPs in the intronic region of the gene EPAS1 (encoding for HIF2 α), has been positively associated with the risk for renal cell carcinoma, yet how HIF2 α is regulated to sustain ccRCC is poorly understood. In this manuscript Naas et al have used epigenomic profiling to identify an enhancer that controls the expression of EPAS1 specifically in ccRCC. The authors show that this enhancer is crucial to the expression of EPAS1, and they suggest that the activity of this enhancer is regulated by HIF and other renal-specific transcription factors. Through this study, the authors have provided a molecular mechanism explaining how the oncogene, HIF2 α , is controlled in ccRCC to contribute to accelerated tumor formation/growth.

Major Concerns:

- Figure1: The authors have shown a representative RNAscope image for EPAS1 in ccRCC tissue (1e). The authors should include images from the other three patients in supplementary figures as well. To also ensure that the signal is specific, the authors should consider including a negative control (either normal tissue or control PTC cells). If the RNAscope probes have been validated in a different study, the authors should include the reference. Could the authors also elaborate on how cancer cells were distinguished from noncancer cells in 1e? Why is the EPAS1 signal restricted only to some areas within the tissue and is this seen across patients?

We thank the reviewer for evaluating our manuscript. In the revised version, we have included images of RNAscope experiments from tissues of additional donors (Supplementary Fig. 2 and 3). We performed RNAscope for HIF-2 α on non-tumor kidney tissue and used a control probe as negative control. The commercially available probe had been validated by an independent group and we have added the citation to the method section (Varberg, K.M. et al. 2023, Nat Commun, <https://doi.org/10.1038/s41467-023-40424-5>).

We had used staining for CD31 or CA9 to identify endothelial or tumor cells, respectively, and have included this data in the new version of the manuscript (Supplementary Fig. 3). As expected from previous analyses (Wiesener M.S., et al. 2002, FASEB Journal, doi.org/10.1096/fj.02-0445fje), endothelial cells displayed high levels of HIF-2 α mRNA in RNAscope experiments. This may explain the somewhat restricted pattern of HIF-2 α mRNA signal observed in the tumor tissue as noted by the reviewer, which appears to be generated by endothelial cells in vessels. However, we also detected substantial HIF-2 α mRNA signals in tumor cells. In these cells, lipids and glycogen are enriched due to underlying metabolic activity of HIF ("clear cell phenotype"), which may lead to accumulation of the cytoplasm signals at cell borders contributing to the pattern of expression. In order to corroborate the finding of increased HIF-2 α mRNA in tumor cells and exclude an incorrect assignment of the HIF-signals, we have analyzed scRNA-seq data and have screened isolated primary tumor cells for HIF-2 α mRNA expression (Fig. 1d and 1g). In both analyses, HIF-2 α mRNA was increased in tumor cells when compared to renal tubule cells.

- Figure 2: The authors have highlighted enh-1031503 as ccRCC-specific. However, this region is also active in PTC from patients 1 and 2, less so in patient 3. Could the authors comment on why this may be? Could it be cross-contamination or some other reason? If the authors perform differential analysis on the K27ac peaks at this region, is there significant difference? The same variability in H3K27ac signal is apparent in Supp Fig 3, when comparing normal and ccRCC tissue. Could the authors elaborate on this variability? Perhaps include the corresponding EPAS1 expression status?

The region of enh-1031503 has been described as a “ccRCC-gained enhancer” by Yao et al. integrating H3K27ac ChIP-seq signals from ten tumors and corresponding normal kidney tissue (Yao, X. et al. 2017, *Cancer Discov*, doi: 10.1158/2159-8290.CD-17-0375). We prefer to use the term “ccRCC-activated” enhancer rather than ccRCC-specific, because the reviewer is right that some accessibility and activity could be seen in tubule cells, especially within the upstream region which interacted with the lineage-specific factors PAX8 and HNF-1 β . However, activity of the whole region increased substantially in ccRCC cells. We speculate that HNF-1 β and PAX8 bound to this region in tubule cells explaining some of the “basal” activity. In contrast, the element that interacted with HIF (KIRC_10770) showed a much more prominent increase in accessibility indicating that HIF contributed to some extent to the gained activity. As outlined below and in the new version of the manuscript, we performed a great number of additional epigenetic analyses using transcription factor knock-out or enhancer-defective cells which confirm this hypothesis.

We performed analyses of differential activity for this region as requested by the reviewer. Although activity was increased at this site in tumor cells of every of the three individuals compared to corresponding tubule cells, the effect was not significant in the combined analysis probably because of the low number of samples ($p=0.14$, Point-by-point Fig. 1). In the H3K27ac data from Yao et al., which comprised ten tumors and corresponding normal tissue, we detected a positive correlation of HIF-2 α mRNA expression and H3K27ac ChIP-seq signal at the *EPAS1*-enhancer E2. This result just missed significance in a Spearman’s rank correlation analysis ($R=0.62$; $p=0.06$). We provide this data now in Supplementary Fig. 7c. Of note, in the KIRC ATAC-seq data the positive correlation of *EPAS1* expression with accessibility of KIRC_10770 was significant (Fig. 2d).

Point-by-point Figure 1

Point-by-point Figure 1: Activity of the ccRCC-activated *EPAS1*-enhancer. **a)** Tracks of H3K27ac ChIP-seq experiments performed on chromatin from primary tubule (PTC) and tumor (ccRCC) cells from patients #1, #2 and #3 at the *EPAS1* locus. The ccRCC-enhancer comprises a region with increased H3K27ac signal in tumor cells in comparison to tubule cells. Data is similar to Fig. 2a and Supplementary Fig. 6a. **b)** and **c)** Quantification of the H3K27ac signal at the ccRCC-gained enhancer (enh-1031503) and a positive control HIF-binding site (*NDRG1* enhancer) for PTC or ccRCC cells from data shown in a). Signals were normalized to reads per genomic content (RPGC) and the integral for the respective locus was calculated. Please see Method section for details. Two-tailed unpaired t-test.

We can exclude cross-contamination of tubule cells with tumor cells for various reasons: a) we use tissue specimens for isolating PTC with great distance to the renal tumor; b) we routinely check cell morphology in our cultures; c) we did not detect stabilization of HIF in untreated PTC in immunoblots (Fig. 1f and Supplementary Fig. 4a) or mRNA signals of HIF activation in RNA-seq experiments; d) we checked RNA-seq data from the primary tubule cells which were generated in parallel to the epigenetic data for reads with *VHL*-mutations corresponding to those in the respective tumor from the same patient. For patient #1, we detected a mutation within exon 2 in the tumor cells (c.343C>A) which was absent in any of the reads of PTC RNA-seq from the same patient (Point-by-point Fig. 2). Thus, we can rule out cross-contamination of the signals from PTC with tumor cells. Regarding activity of this region in non-tumor cells, we would like to point out that H3K27ac signals at this site were also detected to some extent in normal tissue in the data generated by Yao et al. (Yao, X. et al. 2017, Cancer Discov, doi: 10.1158/2159-8290.CD-17-0375, Supplementary Fig. 7a) indicating some “basal” activity in non-tumor tissue as discussed above.

Point-by-point Figure 2

Point-by-point Figure 2: ccRCC cells depict specific *VHL* mutation not detectable in corresponding primary tubule cells. RNA-seq tracks (BAM format) from lysates from primary tubule (PTC) and tumor (ccRCC) cells from patient #1 at exon 2 of the *VHL* gene. Technical duplicates (replicate #1 and #2, respectively) were sequenced. The mutation c.343C>A (green bar in ccRCC RNA-seq) is present in all reads from the tumor cells, but not in PTC reads. Please also note the substantially decreased absolute read number across this exon of *VHL* in ccRCC compared to PTC.

In Fig. 2e, the authors should also include H3K27ac signal, wherever possible, to compare the degree of enhancer activity across these tumor types.

The data presented in this figure (now Fig. 2c) was derived from a set of ATAC-seq experiments in a subset of tumors from the TCGA collection (Corces, M.R. et al. 2018, Science, doi: 10.1126/science.aav1898). Unfortunately, there is no corresponding data set available for all of these tumors for H3K27ac. However, we now provide a detailed analysis of the H3K27ac signals at this site for the Yao et al. data including normal and tumor tissue (please see comment above, Fig. 2b, Supplemental Fig. 7, Yao, X. et al. 2017, Cancer Discov, doi: 10.1158/2159-8290.CD-17-0375).

- Figure 3: When the authors knock-out HIF1 β (3f, g), there is a modest decrease on HIF2 α . Is this modest decrease in line with what is expected? Is this true even at the EPAS1 transcript level? The authors have shown how ATAC signal changes at the ccRCC-enhancer after re-introducing VHL (Fig 3e). The authors should also depict how the enhancer activity is changing as VHL is being re-expressed. Additionally, the authors show how EPAS1 RNA levels change upon addition of DMOG. Can the authors include how the EPAS1 enhancer status is changing in this case?

Thank you for these comments. Yes, we would expect only a modest decrease in target gene expression when manipulating the activity of a distant HIF-binding enhancer. For example, in previous work we reduced expression of MYC mRNA by approx. 30% when inactivating a distal HIF-binding enhancer element in ccRCC cells (Grampp, S. et al. 2016, Nat Commun, doi: 10.1038/ncomms13183). In a recent study, authors inactivated the HIF-binding regulatory element which is involved in ccRCC-specific *CCND1* expression and thereby reduced *CCND1* mRNA levels by approx. 60% in ccRCC cells (Patel, S.A. et al. 2022 Nature, <https://doi.org/10.1038/s41586-022-04809-8>). Thus, the size of the effects measured for the *EPAS1*-enhancers on *EPAS1* transcription was in the same order of magnitude as determined for other important HIF-binding ccRCC enhancers. This is also in line with previous findings that HIF-stabilization increased expression of already transcribed genes rather than leading to *de novo* expression (Xia, X. and Kung, A.L. 2009, Genom Biol, doi: 10.1186/gb-2009-10-10-r113).

We added mRNA expression data for HIF-2 α to the HIF-1 β ko experiments in 786-0 and RCC4 cells as requested (Fig. 3e and 3f). HIF-2 α mRNA was decreased upon HIF-1 β ko in both cell lines.

We added H3K27ac CUT&Tag-seq to the set of experiments in which we re-introduced VHL or knocked-out HIF-1 β (Fig. 3j). Levels of H3K27ac decreased at KIRC_10770 upon reduction of HIF-levels.

In line with absent HIF-binding under DMOG conditions (Fig. 4c), we did not observe increased chromatin activity at KIRC_10770 as measured by H3K27ac ChIP-seq in DMOG treated PTC. We provide this data for the reviewer below (Point-by-point Fig. 3). This work is part of a different study on HIF-activation in tubule cells for which we are in the process of preparing a manuscript.

Point-by-point Figure 3

Point-by-point Figure 3: Chromatin activity at the ccRCC-activated *EPAS1*-enhancer 2 is not changed in primary tubule cells following to HIF-stabilization by DMOG. **a)** Tracks of H3K27ac ChIP-seq experiments from chromatin of primary tubule cells (PTC) exposed to 1mM DMOG for 4h or control conditions. The ccRCC-enhancer *enh-1031503* is highlighted in yellow. **b)** and **c)** Quantification of the H3K27ac signal at the ccRCC-gained enhancer (*enh-1031503*) and the positive control locus *NDRG1* enhancer as a known HIF-binding site for PTC in samples shown in a). Signals were normalized to reads per genomic content (RPGC) and the integral for the respective locus was calculated. Two-tailed unpaired t-test.

- Figure 4: It is unclear if the HIF ChIP-seq signal in 4a is from WT/untreated PTC cells or cells treated with DMOG. Can the authors point out where exactly KIRC_10770 and KIRC_10767-69 lie? Are the HIF peaks specifically on KIRC_10770? Is there some difference in motifs or methylation status at enhancer 1 versus enhancer 2 that could explain why enhancer 1 shows stable HIF binding while enhancer 2 shows more dynamic binding? Based on Fig 4d, it appears as though PAX8 and HNF1 β peaks do not coincide on KIRC_10770. What is the level of change in H3K27ac levels when these TFs are knocked-out in 786-O cells? Can the authors include this data to strengthen their claim that PAX8 and HNF1 β regulate HIF2 α , and that enhancer 1 or 2 maybe involved?

The ChIP-seq tracks for HIF-binding in PTC as presented in Figure 4a (now Fig. 4c) were generated from DMOG treated cells. We added this information to the figure legend and apologize for the lack of information in the legend in the initial manuscript. HIF signals were detectable at the *EPAS1*-enhancer 1 confirming the presence of a HIF-stabilizing agent. We added detailed illustrations of the different KIRC elements at the *EPAS1*-enhancers to the revised manuscript as requested by the reviewer (e.g. Fig. 2a, Fig. 5 f, Fig. 6a). Yes, HIF peaks within the *EPAS1*-enhancer E2 were specific to KIRC_10770.

We thank the reviewer for the question regarding the methylation status, which is indeed very interesting, since methylation of HREs has been reported to avoid HIF-DNA interactions (D'Anna, F. et al. 2020, Genom Biol, doi: 10.1186/s13059-020-02087-z). To gain first insights into effects of methylation on enhancer function, we resorted to available methylation data of the KIRC tumors and corresponding normal kidney tissue from TCGA. This revealed that methylation of CpGs in the vicinity of both loci was significantly decreased in the ccRCC tumors when compared to normal kidney tissue. In fact, the methylation status of other CpGs in this region was more or less unaffected except for one

other site. However, the methylation status in normal tissue was not different between the CpGs in vicinity of the two enhancers and was more reduced at *EPAS1*-enhancer E1 in the tumors (Supplemental Fig. 15). This may suggest that methylation status changed for both enhancers. However, it does not necessarily explain the more dynamic accessibility for HIF to *EPAS1*-enhancer E2. One limitation of our analysis is that we did not test the methylation status of the specific HREs within the enhancers.

In response to the question regarding the TFs, we performed additional experiments in 786-O cells. First, we generated single clones of 786-O cells with defective PAX8 or HNF-1 β signalling by knocking out these TFs. We then profiled epigenetics of these cells by performing ATAC-seq and H3K27ac CUT&Tag-seq. In differentially accessible DNA-elements defined by ATAC-seq, we noticed a strong enrichment of the respective binding motifs of the two transcription factors (Supplementary Fig. 18). At the *EPAS1*-enhancer E2, we detected substantially reduced accessibility and activity upon depletion of both factors. We added this data to Fig. 5e. Thus, together with the finding that knock-out of HNF-1 β and PAX8 reduced HIF-2 α mRNA (Fig. 5b) the newly generated epigenetic data fully confirmed that PAX8 and HNF-1 β were crucially involved in the regulatory circuitry of *EPAS1* expression via *EPAS1*-enhancer E2. Furthermore, we also confirmed the contribution of the specific binding sites of these TFs within *EPAS1*-enhancer 2 by targeting the binding sites using CRISPR/Cas9 technology. This resulted in reduced HIF-2 α mRNA and protein levels (please see also below and Fig. 5f-h).

- Figure 5: The authors have deleted E1 or/and E2 to assay for the downstream effect on *EPAS1* expression. However, there is insufficient data provided to show that the enhancers have been effectively abrogated. Can the authors include genotyping and DNA sequencing data to show that the HIF motifs are deleted? Can they also include H3K27ac data to show the extent to which the enhancer activity has been affected upon CRISPR-manipulation? It is curious that even when E1+E2 are deleted, HIF2 α is still expressed. Could the authors speculate on why that might be? It is unclear to me, what the authors mean by enhancer 2 activity in trans. Do the authors mean to assess the effect of deleting enhancer 2 on HIF-2 α downstream gene targets? Or did the authors only delete the enhancers on one of the alleles? Are 786-O cells diploid for the *EPAS1* locus? If no, how many alleles for E1 and E2 were successfully deleted by the authors? Could the authors include these details? Although it is mentioned in the methods, in the figure and figure legend, the authors state that the CUT&TAG was performed to assess HIF1 β binding.

We now provide DNA sequencing data (ICE-analysis and sequencing of plasmids carrying the mutated sequences, Supplementary Fig. 20, 22, 23 and 24) to show that CRISPR/Cas9 technology introduced mutations at the targeted sites (e.g. HREs in the HIF binding site). We also performed new H3K27ac CUT&Tag-seq experiments in *EPAS1*-enhancers E2 (E2_HIF) ko cells to explore changes in enhancer activity as requested (Fig. 7f). This revealed reduced activity at the KIRC_10770 element by preventing HIF-interactions.

Regarding the effect size on HIF-2 α mRNA reduction after knocking-out two enhancers we would like to refer to our statement above that the HIF-binding *EPAS1*-enhancers E1 and E2 rather modulate than initiate HIF-2 α expression of target genes similar to other HIF-binding enhancers.

Activity of the enhancer in trans: the reviewer is right, in the initial manuscript we referred to effects of the *EPAS1*-enhancer E2 that were mediated by differential regulation of HIF-2 α , i.e. downstream regulation of HIF-2 targets. To increase clarity and to avoid any confusion with allelic effects, we have changed the wording from “effects in trans” to “effects on downstream HIF-2 targets”. We provide evidence for this effect in Fig. 7e-h. HIF-2 targets were downregulated in cells with a defective HIF-binding site at *EPAS1*-enhancer 2 as measured by RNA-seq.

786-0 cells are hypertriploid. We confirmed CRISPR/Cas9 efficacy for the different sgRNAs by ICE analysis and cloned DNA fragments from *EPAS1*-enhancer ko clones of cells. We detected 2-3 different mutations at the HIF binding sites per clone of cells suggesting that at least two alleles had been affected by CRISPR/Cas9 modulation. In reporter assays, we confirmed that the sequences with deleted HREs had much lower luciferase activity compared to the wild-type sequence when transfected Kelly cells were stimulated with DMOG. We have included this information in Supplementary Fig. 24. Please note, we did not detect residual HIF-binding to *EPAS1*-enhancer 2 by HIF-1 β CUT&Tag-seq confirming that the mutations were sufficient to prevent HIF-binding (Fig. 7f).

We have indicated when HIF-1 β binding was assessed by ChIP-seq or Cut&Tag, respectively, throughout the manuscript.

- Based on the ATAC-seq data, the element KIRC_10770 appears to be the most dynamic. What is the direct contribution of this element in regulating *EPAS1* expression levels? In the CRISPR deletions performed in Fig 5, it is unclear if this element was abrogated or not. The authors should consider specifically deleting KIRC_10770 to assay for downstream effects. Also, does this element carry motifs for HIF? Or for Pax8/HNF1 β ? Surprisingly, contrary to the data from this manuscript, analysis in Supp 6b shows that KIRC_10770 is very modest DNase hypersensitivity in the renal/cancer category. Is it possible for the authors regroup this data to separate out cancer v/s non-cancer samples?

The element KIRC_10770 interacted with HIF in our ChIP-seq and CUT&Tag experiments. As pointed out by the reviewer, accessibility to this element was highest in VHL-defective renal cancer cells. This element carries three HREs. We had targeted two of these HREs in the CRISPR/Cas9 experiments (sgRNA E2_HIF in Fig. 6c-i and Fig. 7), which led to mutations abolishing HIF-binding (Fig. 7f) and led to decreased HIF-2 α mRNA levels in different cell lines (Fig. 6c, d and 7d). We provide data for the mutations introduced to this site and analysis of reporter activity of vectors harbouring the enhancer sequence with the different deletions in Supplementary Fig. 22-24.

We agree that DNase hypersensitivity at this site in the renal/cancer category from the 733 biosample data set was modest. However, it was the only category that displayed substantial accessibility across the tissues and this finding corresponded well to the data presented for KIRC_10770 in the TCGA ATAC-seq analyses of KIRC and the other tumors (Fig. 2c). The different categories of accessibility in Meuleman et al. (Meuleman, W. et al. 2020, Nature, doi.org/10.1038/s41586-020-2559-3) were derived from a subset of the 733 biosamples. The renal/cancer category consists of 89 samples, most of them are from fetal renal tissue (74 samples) which did not show accessibility to KIRC_10770. However, we identified the selection of adult renal cancer as well as renal tubule derived samples and provide the data below (Point-by-point Fig. 4). Albeit the number of samples is low, this analysis confirmed increased accessibility in renal cancer samples. Thus, this provides further evidence for tissue specificity and functionality of KIRC_10770.

Point-by-point Figure 4

Point-by-point Figure 4: Chromatin accessibility at the ATAC element KIRC_10770 is increased in renal cancer cell samples compared to tubule derived samples. ATAC site KIRC_10770 overlaps with the chromatin region chr2:46233000-46233180 defined by DNase I hypersensitivity from the 733 biosamples set published by Meuleman et al. Only samples derived from adult tissue or cell lines in the compartment “renal/cancer” are shown.

Minor Concerns:

- The authors can include quantification for the western blots to depict the degree of change in HIF2 α levels upon the different perturbations throughout the manuscript.

We quantified western blots and provide data for this in a separate supplemental table (Supplementary Table 12). In general, effects caused by the different treatments on HIF-2 α mRNA levels were consistent with effects measured on HIF-2 α protein in immunoblots (on average approx. 50% reduction across all conditions tested). Evidence that these changes in protein levels were also functional came from the RNA-seq experiments in which direct downstream HIF-2 targets displayed reduced expression upon *EPAS1*-enhancer manipulation.

- Fid 1d is not necessarily adding to the figure and can be removed or moved to supplementary.

We have removed this figure.

Reviewer #2, expertise in ccRCC, epigenetics and hypoxia (Remarks to the Author):

HIF-2a is one of the most important driving factors in clear cell renal cell carcinoma. Naas et al. present a very interesting manuscript that identifies a feed-forward regulatory circuit controlling HIF-2a (*EPAS1*) mRNA expression in ccRCC. This involves an enhancer element that regulates HIF-2a mRNA expression.

The study makes excellent use of human cell resources from normal kidney tissue and from ccRCCs, of cultured ccRCC and other cancer cell lines, conducts different analyses of chromatin structure, status and protein binding (ATAC-seq, ChIP-seq, CUT&TAG), combined with RNA expression analyses and functional mutation of identified genomic sites as well as of the transcription factors in the regulatory network. The study also integrates many different publicly available datasets from previous studies to support the experimental work (however, many of the figures are in fact entirely or heavily based on already published data). While the data presented in the manuscript suggest that the overall mechanism that the authors have uncovered is very likely to be true, there are several aspects of the study that remain incomplete to fully support the proposed model and conclusions, as well as to address some important open questions that are central to the overall findings. Most of these could be readily addressed using the tools that the authors describe in the manuscript.

Major issues

1. A very interesting observation of the overall study is that the HIF-2a enhancer E2 appears to be relatively specific to ccRCC (and also to glioblastoma) which might explain in part the sensitivity of the kidney to the oncogenic effects of HIF-2a. However, there are several limitations to the current study that prevent this conclusion being made. Fig 3b-e and Suppl Fig 8 nicely show that VHL re-introduction into ccRCC cells suppresses HIF protein abundance, HIF-2a mRNA abundance and loss of ATAC-seq signal at the enhancer, showing that there is a dependency on VHL in established cancer cell lines. However, efforts to show the dependency of this activity on HIF fall short in their current form. Experiments in Fig 3f and g would need to be complemented by HIF-2a mRNA measurements and crucially by ATAC-seq to investigate the status of the enhancer. Enhancer status assessment is also needed for experiments in Fig 4f and g which investigate the effects of mutation of PAX8 and HNF-1b. It remains unclear whether PAX8, HNF-1b or HIF factor binding is mutually dependent and what effect these factors have on the accessibility of the enhancer.

We thank the reviewer for evaluating our manuscript and for the constructive comments. In order to complement our data on enhancer function, activity and interactions, we performed a series of new experiments along the suggestions of this referee and referees 1 and 3. In the revised version, we have included ATAC-seq, H3K27ac CUT&Tag-seq and mRNA data from cells with transcription factor knock-out (HIF-1 β , HNF-1 β , PAX8) or defective transcription factor-binding elements (Fig. 3, 5, 7). Confirming a regulatory effect of these factors and the DNA-elements on HIF-2 α expression, the CRISPR/Cas9 engineered cells displayed reduced accessibility and activity of the corresponding enhancer elements. HIF-2 α mRNA and protein were also significantly reduced when compared to control treated cells.

Regarding the dependency of *EPAS1*-enhancer E2 activity and *EPAS1* regulation on HIF, we now can provide multiple lines of evidence. First, HIF-1 β knock-out, which prevents HIF- α DNA interactions at HRE containing sequences, led to reduced HIF-2 α mRNA and protein in 786-0 and RCC4 ccRCC cells (Figure 3 e-h). Second, exposure of pVHL re-expressing RCC4 cells to the HIF-stabilizer DMOG led to increased HIF-2 α mRNA levels (Supplementary Fig. 10). Third, in primary tubule cells siRNA mediated

knock-down of HIF-1 α or HIF-1 β prevented DMOG-induced increases of HIF-2 α mRNA (Figure 4b). Fourth, targeting HREs within *EPAS1*-enhancer 2 in ccRCC cells abolished HIF-binding to this site and led to diminished HIF-2 α mRNA levels (Fig. 7d, f).

In the settings of HIF-1 β knock-out, DMOG-exposure of VHL-re-expressing RCC4 cells as well as HRE-targeting in ccRCC cells, we conducted new ATAC-seq and H3K27ac CUT&Tag-seq experiments and have included this data in the revised manuscript (Fig. 3i, j, Fig. 7f, Supplementary Fig. 10). These experiments confirmed reduced accessibility and activity at *EPAS1*-enhancer E2 upon the manoeuvres mentioned above.

We also explored dependency of HIF-binding to KIRC_10770 on the presence of the lineage-specific factors PAX8 and HNF-1 β . For this, we conducted HIF-1 β CUT&Tag-seq experiments in 786-0 cells depleted for PAX8 or HNF-1 β or mutated at the binding sites of both factors at *EPAS1*-enhancer E2. In all conditions, we detected reasonably preserved HIF-1 β signals at the KIRC_10770 element irrespective of the presence of either of the two lineage-specific TF within the cell or at the enhancer. This indicates that HIF-binding to KIRC_10770 was not entirely dependent on PAX8 or HNF-1 β (Supplementary Fig. 21).

2. The overall question which needs to be addressed is how the enhancer becomes accessible in ccRCC. It goes from a state of inaccessibility in normal renal proximal tubule epithelial cells to being accessible in ccRCC cells. Based on the presented evidence it appears likely that this transition might be (at least partly) induced by loss of VHL and be HIF-dependent, however other genetic/epigenetic alterations that have arisen in these cancer cell lines might have also cooperated with VHL loss to induce this state (given the very long timeframe of ccRCC evolution this is plausible). This concept should be investigated. Is VHL loss and HIF- α activation necessary and/or sufficient for enhancer activation? For example, it should be investigated if mutation of VHL or overexpression of non-degradable HIF-1 α and HIF-2 α mutants in normal renal epithelial cells alters endogenous HIF-2 α mRNA levels and enhancer status?

It would also be highly interesting to conduct hypoxia time course experiments in normal renal cells as hypoxia has previously been shown to induce a transient HIF-1 α increase, followed by a longer-term HIF-2 α increase. It may be that the feed-forward enhancer mechanism identified here plays a role in this effect.

We fully agree with the reviewer that the mechanism(s) which change(s) enhancer accessibility and activity are of outstanding interest to better understand malign transformation in ccRCC. Regarding HIF-binding enhancers, a very similar observation of differential activity between tubule and tumor cells has been made for the *CCND1* enhancer, which appears to be specifically accessible in ccRCC cells (Schödel, J et al. 2012, Nat Genet, doi: 10.1038/ng.2204 and Patel, S.A. et al. 2022, Nature, <https://doi.org/10.1038/s41586-022-04809-8>). As pointed out by the reviewer, the process of changing chromatin accessibility might take years or decades and might also involve additional mechanisms/mutations apart from VHL-inactivation. It is also possible that only a small cell population of the tubular system is susceptible to malign transformation and that this population is epigenetically predisposed to allow HIF-interactions at oncogenic enhancers as early as VHL is lost (e.g. *CCND1*, *EPAS1*). Identifying these mechanisms or the cell population is ongoing work, but we feel out of the scope for this manuscript. However, to begin to address this question we performed a series of experiments in different cellular backgrounds and analysed existing data.

For example, we inactivated *VHL* in immortalized PTC derived from the urine of VHL-patients (huPTC) who are especially predisposed to develop renal cancer and performed RNA-seq and ATAC-seq on

single clones of these cells after approx. 14 weeks of culture. Confirming the loss of functional pVHL, we did detect stabilization of both HIF- α isoforms upon VHL-knock-out in immunoblot analyses in huPTC (Point-by-point Fig. 5a). However, RNA-seq did not reveal increases of HIF-2 α mRNA and accessibility to the HIF-binding element KIRC_10770 was not different between control and VHL-knock-out cells (Point-by-point Fig. 5 b, c).

We also identified an ATAC-seq data set published by Patel et al. (Patel, S.A. et al. 2022 Nature, <https://doi.org/10.1038/s41586-022-04809-8>) which was derived from the HK2 immortalized proximal tubule cell line overexpressing HIF-2 α or empty vector. Patel et al. reported no change of accessibility at the *CCND1* HIF-binding enhancer after 6 weeks of culture of these cells. Similarly, we did not detect changes in accessibility to KIRC_10770 when inspecting this element in the published data (Point-by-point Fig. 5 d). However, these cells are capable of inducing HIF-2 α mRNA upon HIF-stimulation by DMOG probably via binding to *EPAS1*-enhancer E1 (Point-by-point Fig. 5 e, f). This indicates that similar to the *CCND1*-enhancer additional mechanisms must operate to render *EPAS1*-enhancer E2 more accessible in full-blown tumor cells.

As requested by the reviewer, we have performed long-time hypoxia exposure in primary tubule cells. These cells tolerate hypoxia at 0.1% O₂ for up to 48 hours. In ATAC experiments, we did not determine differential accessibility to the KIRC_10770 element at *EPAS1*-enhancer (E2_HIF) in the hypoxic cells compared to control conditions and negative control loci. Of note, accessibility to *EPAS1*-enhancer E1 increased substantially under hypoxic conditions comparable to the accessibility at a positive control locus at the *NDRG1* gene (Point-by-point Fig. 5 g-i). We show data from two PTC isolates and 786-0 cells (cultured in normoxia) as a control. As mentioned above identifying factors that increase accessibility of KIRC_10770 is ongoing work.

Taken together, *VHL*-knock out in huPTC from a VHL-patient, overexpression of HIF-2 α in the immortalized proximal tubule HK2 cell line or long-term hypoxia in primary tubule cells did not change accessibility to KIRC_10770. Thus, we speculate that additional mechanisms must become active to release HIF-2 α regulation from this element. We are currently following this up in a separate study by performing genetic manoeuvres in addition to *VHL*-knock out in the primary cells.

Point-by-point Figure 5

Point-by-point Figure 5: Accessibility to KIRC_10770 within *EPAS1*-enhancer E2 cannot be induced by loss of pVHL, HIF-2 overexpression or hypoxia in primary tubule cells. **a**) Immunoblot analyses for HIF-2 α , HIF-1 α , pVHL and β -actin in lysates from single clones of human urine-derived primary tubule cells (huPTC). Cells were donated from a patient with VHL disease (het *VHL* loss). CRISPR/Cas-mediated complete knock-out of *VHL* (hom *VHL* loss) leads to stabilization of HIF isoforms. **b**) Volcano plot depicting differentially expressed genes (DEG) in RNA-seq experiments using single clones of huPTC shown in a). Each RNA sample was sequenced in technical duplicates. Yaxis indicates $-\log_{10}$ adjusted p-value as calculated by the Benjamini-Hochberg approach in DESeq2. **c**) ATAC-seq tracks for single clones of huPTC with heterozygous (het) and homozygous (hom) loss of VHL at the *EPAS1* (left) and *NDRG1* (right) locus. ATAC-element KIRC_10770 is highlighted in orange. HIF-1 β ChIP-seq track indicates HIF- β binding sites in 786-0 cells for comparison. **d**) ATAC-seq tracks from lysates of HK2 cells transfected with empty vector (EV) or VHL-resistant HIF-2 α protein (HIF2A) at the *EPAS1* (left) and *NDRG1* (right) locus. Data was published by Patel et al. (GEO: GSE163485) **e**) and **f**) Expression qPCR analyses for HIF-2 α mRNA (**e**) or EGLN3 mRNA (**f**) in lysates from HK2 exposed to 1mM DMOG or control conditions for 16h. **, *p* < 0.01. Mean + SD. One sample t-test with a hypothetical value of 1. *n* = 6 independent experiments. **g**) and **h**) ATAC-qPCR for E2_HIF (KIRC_10770) and negative as well as positive control loci (E1 and NDRG1_HIF) in PTC from two different patients. Cells were exposed to hypoxia (0.1% O₂) for 24h or 48h or cultured in normoxia. Values were normalized to a control site in proximity of *EPAS1*-enhancer 2 depicting no relevant accessibility in ATAC-seq data of the respective cell type. **i**) ATAC-qPCR for E2_HIF and negative as well as positive control loci (E1 and NDRG1_HIF) in 786-0 cells.

3. Fig 3h and 3i lack evidence that DMOG was active in these cells (e.g. HIF- α western blots or activation of well characterised HIF-target genes) and proof of functional knockdowns is lacking.

We have included HIF-1 α western blots as well as quantification of HIF target gene expression by qPCR from a selection of the primary cell cultures to give proof of HIF-stabilization by DMOG (Supplementary Fig. 11 and 12). We also performed immunoblots for HIF and qPCR for *EGLN3* expression using PTC upon HIF-knockdown experiments (Supplementary Fig. 12c, d).

4. Fig. 3b, 3c. The argument presented around these figures is not convincing as it relies on the absence of an effect to prove a hypothesis. There is such a massive spread of expression levels in TCGA data across each of the three genotypes that it appears very unconvincing that there is any relationship between these genotypes and HIF-2 α mRNA.

Thank you for this critical comment. We had presented the eQTL data derived from PanCanQTL analyses (Gong, J. et al. 2018, *Nucleic Acids Res*, doi: 10.1093/nar/gkx861) in the first version of the manuscript because it highlighted the region harboring the *EPAS1*-enhancer E2 as being linked to HIF-2 α expression especially in the KIRC data compared to the other tumor entities. However, we agree that the data presented for genotype-dependent HIF-2 α expression in PTC did not prove the postulated hypothesis of differential action of the genotype when comparing tumor vs primary cells. Thus, we have removed the genotype-expression data and the corresponding paragraph from the manuscript and only kept the track with the published PanCanQTL SNPs in new Figure 4c.

In the context of interactions of the *EPAS1*-enhancers E1 and E2 with genetic susceptibility to renal cancer and also in response to a comment from referee 3, we accessed the KIRC genotype and mRNA expression data to test intronic variants which are associated with the development of renal cancer for effects on HIF-2 α expression. For SNP rs11894252 which was reported in the initial GWAS (Purdue, M. et al. 2010, *Nat Gen* doi: 10.1038/ng.723) and which was available for analysis on the genotype array used by TCGA, we did not detect an eQTL for HIF-2 α mRNA (Supplementary Fig. 17b), which is in line with a previous report (Lasker, R.S. et al. 2019, *Eur J Hum Genet.*, doi.org/10.1038/s41431-019-0455-9). However, we also evaluated rs12617313, which had been detected as an independent signal in a follow-up fine-mapping study and for which the risk allele was strongly associated with the presence of *VHL*-mutations in tumors (Han, S.S. et al. 2012, *Human Molecular Genetics*, doi.org/10.1093/hmg/ddr551). This SNP is a significant eQTL for HIF-2 α mRNA expression in the TCGA KIRC tumor tissues, but not in non-diseased kidney tissue (Fig. 4d). We agree that there is a wide spread in this kind of data, but effects of common variants in non-protein coding regions on gene expression are small in general. We detected a clear genotype-dependent expression in this analysis. Moreover, using our large cohort of PTC which we genotyped for this SNP, we observed a similar association of the rs12617313 genotype with HIF-2 α expression (Fig. 4e). Importantly, this association was only detectable in cells in which HIF had been stabilized by DMOG, but not in untreated cells. A recent study (Schmid, V. et al., 2019, *Sci Rep*, doi: 10.1038/s41598-019-55098-7) also confirmed that this intronic region physically interacts with the upstream *EPAS1*-enhancer E1 indicating the existence of a complex interplay of enhancers with intragenic regions of *EPAS1* in renal cancer. Thus, we can now link HIF-2 α expression with effects of ccRCC-associated germline variants and HIF-activated enhancers. This is the first report that explains some of the genetic associations of the intronic genetic variants within *EPAS1* in *VHL*-defective ccRCC biology.

Since the effect appears to be dependent on loss of *VHL* and/or HIF-activation, this puts this genetic locus in line with or – since HIF-2 is one of the master upstream regulators – even ahead of other important RCC risk loci interacting with HIF such as *CCND1* (Schödel, J. et al. 2012, *Nat Genet*, doi:

10.1038/ng.2204 and Patel, S.A. et al. 2022, Nature, <https://doi.org/10.1038/s41586-022-04809-8>) or *MYC* (Grampp, S. et al. 2016, Nat Commun, doi: 10.1038/ncomms13183). We have included these novel findings in Fig. 4d, e and Supplementary Fig. 17.

5. Fig 5. The effects of the sgRNAs at the target loci in whole populations and single cell clones should be verified by DNA sequencing.

We routinely perform ICE-analysis from Sanger sequencing data as well as sequencing of plasmids carrying the introduced mutations to verify effects of sgRNAs experiments. We provide representative examples of this data for the CIRSP/Cas9 experiments conducted in this study in Supplementary Fig. 20, 22-24.

6. Supplementary Figure 6d is difficult to understand and currently not possible to interpret. A diagram of the vectors that are being tested would be helpful. It is not clear what normalized luciferase intensity is referring to as there is no mention of the empty luciferase vector. From this it appears that the normalization is simply a transfection normalization (beta-galactosidase) but luciferase signals are not relative to an empty vector. It is therefore impossible to conclude whether DMOG treatment has any specific effects on the E1 and E2 elements or whether this is a non-specific effect on the luciferase plasmid in general. Cloning HRE mutants into the E1 and E2 elements would provide important additional evidence that the E1 and E2 elements are responsive to HIFs.

We apologize for the lack of clarity regarding strategy and normalization of reporter assays. We have included a diagram of the vector used in new Supplementary Fig. 14e. In the revised figure, we have included the data from empty vector controls for experiments testing the HIF-binding sequences from *EPAS1*-enhancers E1 and E2 upon HIF-stabilization in reporter assays (Supplementary Fig. 14d). We amended the figure legend to clarify how normalization was performed: "Values of luciferase activity were first normalized to activity of co-transfected β -galactosidase and subsequently to the activity of the respective control condition without HIF-stabilization". As requested by the referee, we also generated reporter constructs with different HRE mutations at KIRC_10770 introduced by the sgRNAs and performed reporter assays using DMOG as HIF-stabilizing agent (Supplementary Fig. 24). This confirmed reduced reporter gene activation by DMOG in cells transfected with HRE mutant vectors indicating functionality of the HREs.

7. Overall the study lacks in vivo evidence. Figure 6 essentially represents a series of reanalyses of published data (b,c,d) that correlate with the model that is being proposed by the current study. 6e and f show a cell culture 3D growth assay presumably as a surrogate for a xenograft assay comparing control clones and E2 ko clones but there is no evidence that the observed growth differences are dependent on the enhancer or on changed HIF-2a expression levels. A rescue experiment with exogenous HIF-2a expression would be beneficial to support the dependency of this assay on HIF-2a levels. It is currently unclear what the experiments shown in Fig 6 add compared to the published studies and don't even directly validate the enhancer hits that were identified in the published screen from the Nature paper (reproduced in Fig 6d).

We now have performed tumor xenograft experiments using the two different *EPAS1*-enhancers E2 knock-out (E2_HIF2.1 and 2.2) and two control (nt 0.1 and 0.2) clones of 786-0 cells. Inactivating the HIF-binding site in *EPAS1*-enhancer 2 prevented tumor growth in NOD/SCID-gamma mice (Figure 7b, c, Supplementary Figure 25a). In order to generate these cells, we had used two HRE-targeting sgRNAs

that corresponded exactly to two out of eight sgRNAs employed by Patel et al. in CRISPRi experiments in xenograft models (Patel, S.A. et al. 2022 Nature, <https://doi.org/10.1038/s41586-022-04809-8>). We have clarified the concordance of the two sgRNA sequences from our study with two sgRNAs from Patel et al. in the revised manuscript. We characterized the *EPAS1*-enhancers E2 knock-out cells in this work now extensively and provide evidence for clear HIF-2 driven downstream effects. We removed the 3D growth assay from the manuscript. Thus, our work complements the wealth of existing data on the role of HIF-2 α in promoting renal tumor growth.

Regarding presentation of published data, we have removed former Fig. 6 b and c from the manuscript. We refer to these results showing higher sensitivity to HIF-2 inhibition in tumors with high levels of HIF-2 α mRNA now in the introduction and the discussion section. We have moved former Fig. 6d to supplement material (Supplementary Fig. 25b). Patel et al. used 8 different sgRNAs (two each in four constructs) to target this region, but in their article they plotted values only for a combined analysis of the two most depleted constructs per region. We had plotted the data from the CRISPRi screen in a different way than Patel et al. so that the effects of all four sgRNAs combinations targeting this region became obvious independently.

Minor issues

1. Some of the diagrams that are presented as figure panels are not necessary to understand these simple experiments. Figs 1d, 1g, 2a, 5f could be removed from the figures.

We removed most of the diagrams as requested except new figure 1e, which we would like to keep in the manuscript, since it illustrates that tumor and tubule cells were derived from the same patient.

2. While many figures use data mining to interrogate datasets generated in the current study, quite a few figure panels are simply re-representations of data generated in other studies so should not be shown again. e.g. the representation of TCGA data shown in Fig 1b has been previously published by Nature Communications in another study (<https://doi.org/10.1038/s41467-020-17873-3>). Fig 6b and 6c are re-representations of published figures in Nature ([doi:10.1038/nature19796](https://doi.org/10.1038/nature19796)) and Fig 6d replots data obtained in a recent Nature publication (<https://doi.org/10.1038/s41586-022-04809-8>).

As requested, we either moved the data to supplemental material (former Fig. 1b and former Fig. 6d) or removed it from the manuscript (former Fig. 6b and c). We refer to the results showing higher sensitivity to HIF-2 inhibition in tumors with high levels of HIF-2 α mRNA in the introduction and the discussion sections (Chen, W. et al. 2016, Nature, doi.org/10.1038/nature19796).

3. The axes of Fig 2d are not labelled in the figure.

Thank you. We have labelled the axes in this figure in the new version.

4. The western blot lanes in Fig 3c should be labelled above the blots as it is not immediately clear that these are the same conditions as in 3d below.

Thank you. We have included separate labelling for immunoblot and mRNA data.

5. It would be helpful for the DNA nucleotide HRE sequences that are being targeted in Fig 5a to be

shown in relation to the sgRNAs that target these sequences so that the reader can understand what type of mutations are being introduced

We now provide a supplementary figure that displays sequence information of the HRE carrying elements and the sgRNA guides used for targeting these elements (Supplementary Fig. 19).

6. Supplementary Fig 6. The authors state that a small fraction of non-ccRCC cell lines show signal at enhancer 2 as further evidence of the ccRCC specificity of enhancer binding of HIF-b. However, 3 of 8 cell lines show this signal so this statement is not supported by evidence

We thank the reviewer and have amended the statement to: “Besides, following HIF-stabilization we also detected HIF-signals at *EPAS1*-enhancer E1 (2 out of 7 cell lines) or at *EPAS1*-enhancer E2 (2 out of 7 cell lines) in non-renal tumor cells as well as in immortalized HKC-8 renal tubule cells and VHL-competent ccRCC Caki-1 cells in data from published and newly generated sequencing experiments. This suggests that some other tumor cells might regulate *EPAS1* expression from these sites (Supplementary Fig. 14a).”

For “2 out of 7 cell lines” we refer to non-renal cells.

Reviewer #3, expertise in kidney cancer and ATAC-seq (Remarks to the Author):

HIF has been known to be important in the development and progression of renal clear cell carcinoma though its mechanism of dysregulation in the cancer is largely unknown. In the article, Naas et al. used epigenome and transcriptome data to study the mechanism of augmented EPAS1 expression in ccRCC. They found that HIF-2 α is highly expressed in ccRCC compared to normal samples. According to the enhancer-gene linkage inference, the ccRCC-specific enhancer enh-1031503 shows great potential to regulate gene EPAS1 expression through the binding of HIF, HNF, and PAX8 transcription factors. They have also verified that knocking out the enhancer can inhibit the development of ccRCC, which indicates a great potential for clinical application for the study. I also have a few comments:

Major comments

1. I understand this is a new mechanism of HIF-2 α regulation, could you also explain the impact and potential application of the study? What is the advantage of blocking E2 instead of eliminating HIF-2 α expression?

We thank the reviewer for these fundamental questions. Accessibility to the *EPAS1*-enhancer E2 correlated with HIF-2 α expression. Thus, an epigenetic screening of tumors for activity of this enhancer could support selection of patients for anti-HIF-2 therapy in the future.

Regarding a potential advantage of blocking the *EPAS1*-enhancer E2 instead of targeting HIF-2 protein, we would like to refer to observations that have been made in some of the tumorgraft ccRCC-models: under treatment with a HIF-2 inhibitor, mutations in *HIF-2 α* or *HIF-1 β* were detected which allowed tumors to escape HIF-2 targeting therapy (Chen, W. et al. 2016, Nature, doi.org/10.1038/nature19796). Thus, in principle blocking HIF-2 α mRNA expression by targeting *EPAS1*-enhancer E2 could represent an attractive alternative therapeutic option to decrease HIF-2 levels in patients with resistance to therapy aiming at HIF-2 protein. Moreover, one could speculate that compared to directly targeting HIF-2 protein adverse effects of this therapy might be not as pronounced, because functionality of the ccRCC-activated element appears to be limited to cancer. For example, reduction of erythropoietin expression and resulting anaemia is the most common adverse event caused by belzutifan therapy in RCC patients (Choueiri, T.K. et al. 2024, N Engl J Med, DOI: 10.1056/NEJMoa2313906). Anaemia occurs in more than 80% of patients. From our data, one would not expect such drastic effects on erythropoietin expression in interstitial cells of the kidney by specifically targeting the tumor DNA-element *EPAS1*-enhancer E2.

2. In vivo and survival studies shall be considered to prove the impact of E2 regarding its impact on cancer development and progression.

In the light of your comment and a comment from referee 2, we have performed tumor xenograft experiments using two different *EPAS1*-enhancers E2 knock-out (E2_HIF 2.1 and 2.2) and two control (nt 0.1 and 0.2) clones of 786-0 cells. Inactivating the HIF-binding site in *EPAS1*-enhancer 2 prevented tumor growth in NOD/SCID-gamma mice (Figure 7b, c) confirming its relevance *in vivo*. Thus, our work complements the wealth of existing data on the role of HIF-2 α in promoting renal tumor cell growth. For example, overexpression of HIF-2 α in ccRCC cells led to increased tumor growth in xenograft models (Kondo, K. et al. 2002, Cancer Cell, doi: 10.1016/s1535-6108(02)00043-0; Raval, R. et al. 2005, Mol Cell Biol, doi: 10.1128/MCB.25.13.5675-5686.2005). In line with this, knock-down of HIF-2 α reduced tumor burden in a xenograft model (Kondo, K. et al. 2003, PLoS Biol, doi:

10.1371/journal.pbio.0000083). More recently, specific pharmacological inhibition of HIF-2 by PT2399 or by systemic administration of anti-HIF-2 α siRNA reduced xenograft growth (Chen, W. et al. 2016, Nature, doi.org/10.1038/nature19796; Cho, H. et al. 2016, Nature, doi: 10.1038/nature19795; Ma, Y. et al. 2022, Clin Cancer Res, doi.org/10.1158/1078-0432.CCR-22-0963).

We agree that future studies should focus on the effects of *EPAS1*-enhancer E2 activity on cancer progression and survival in humans. To do this we currently consider epigenetic analyses of enhancer activity in our large cohort of ccRCC patients in order to identify a prognostic value of this enhancer. This will be an immense task, which we feel is outside of the scope of this initial work describing the existence of the enhancer and its interaction with oncogenic signalling.

3. It seems most conclusions are based on 786-0 cell lines (e.g. ChIP-seq data, KO experiment, and xenograft model), which may be biased. Could you also provide the result of RCC4 to make the conclusion more solid? Also, a ccRCC cell line without VHL loss can be a control.

In the new version of our manuscript, we provide comprehensive transcriptomic and epigenetic data for 786-0 and RCC4 cells as requested by the reviewer. Knock-out of the HIF-binding site at *EPAS1*-enhancer E2 evoked remarkable similar transcriptomic changes in both cell lines including the HIF-2 specific target *CCND1* (please see Fig. 6i).

We also provide epigenetic data from primary RCC cells with defective VHL-protein and have performed additional experiments in the VHL-competent ccRCC cell line Caki1. In this cell line, we detected HIF-DNA interactions at *EPAS1*-enhancers E1 and E2 upon pharmacological HIF stabilization suggesting that in addition to VHL-loss, other mechanisms might be involved in creating access for HIF to the *EPAS1*-enhancer E2 under certain conditions (e.g. in glioblastoma cells or this VHL-competent ccRCC cell line; Supplementary Fig. 14a).

Minor comments

1. Is *EPAS1* expression related to the survival of patients?

This is a very interesting question. We have explored the TCGA KIRC cohort for prognostic effects of HIF-2 α mRNA on patient survival using the GEPIA2 online tool (Tang, Z. et al. 2019, Nucl Acids Res, doi.org/10.1093/nar/gkz430). The analysis revealed that increased HIF-2 α mRNA levels in the tumors corresponded to longer survival of the patients (Point-by-point Fig. 6a). Conformingly, in samples from progressed tumor stages lower levels of HIF-2 α mRNA were detected compared to samples from less advanced tumor stages (Point-by-point Fig. 6c). Similar results were observed and have been described for the ccRCC-specific HIF-2 target gene *CCND1* (Point-by-point Fig. 6b, d and e.g. Wang, Q. et al. 2019, Cancer Medicine, doi.org/10.1002/cam4.2313). Currently, it is unclear to us whether this effect is a true result of reduced HIF-2 regulation/expression in tumor cells in progressed stages or whether this association is caused by effects not directly provoked by the tumor cells (e.g. other cell types contribute to HIF-2 mRNA levels and composition of different cell types and their contribution to expression levels might change during tumor progression). In this respect, it would be interesting to test whether accessibility/activity of the *EPAS1*-enhancer E2 in tumor cells is prognostic. As outlined above, we plan this in future work.

Point-by-point Figure 6

Point-by-point Figure 6: EPAS1/HIF-2 α mRNA expression is associated with overall survival and tumor stage in KIRC. a) Survival analysis based on the expression of EPAS1 RNA in the TCGA-KIRC cohort. Groups were separated in “Low EPAS1” and “High EPAS1” according to the median. Hazards Ratio (HR) was calculated based on Cox PH Model. Dotted line indicates 95% confidence interval. **b)** Same analysis as in a) for CCND1 expression. **c)** Box plot depicting the EPAS1 RNA expression in different cancer stages for the TCGA-KIRC cohort. **d)** Same analysis as in c) for CCND1 expression. Data for a) – d) were accessed and analysed via GEPIA2.

2. When you have single-cell ATAC-seq data for ccRCC, could you also verify if E2 is co-accessible with other regulatory elements such as E1 and promoter of EPAS1?

Thank you. In order to approach this question, we resorted to snATAC-seq data from renal tumors and normal tissue published by Wu et al. (Wu, Y. et al. 2023, Nat Commun, doi.org/10.1038/s41467-023-37211-7). We have used the software Cicero (Pliner, H.A. et al. 2018, Mol Cell, DOI: 10.1016/j.molcel.2018.06.044) to interrogate co-accessible regions within the broader region (500 kb) of EPAS1. This analysis revealed a significant correlation of accessibility of EPAS1-enhancer E2 (chr2:46232855-46233286) with EPAS1-enhancer E1 (chr2:46073018-46074032) in ccRCC cells (co-accessibility score: 0.13). This result indicates that both EPAS1-enhancers can be accessible in the same cell. We have included this finding in the result section. In line with our results from bulk-ATAC-seq, we could not detect a signal for KIRC_10770 in normal epithelial cells. Thus, this prevented the correlation of accessibility of EPAS1-enhancer E2 with EPAS1-enhancer E1 under this condition. We could not detect a positive correlation of enhancer accessibility with the EPAS1 promoter.

3. It also requires data from corresponding patient samples such as Figure 4a.

We performed CUT&Tag experiments in primary ccRCC cells for HIF-1 β and detected robust binding of HIF-1 β to EPAS1-enhancers E1 and E2. We have included this data in Supplementary Fig. 13.

4. It seems HIF, HNF, and PAX8 bind to different regulatory elements in 4d. Do you have any direct experiment evidence for the co-binding / collaboration at E2? What if designing sgRNA to target PAX8 or HNF binding sites at E2?

The reviewer is right. HNF-1 β and PAX8 bind to a different regulatory element than HIF as defined by Corces et al. (KIRC_10767: PAX8, KIRC_10767: HNF-1 β , KIRC_10770: HIF; Corces M.R. et al. 2028, Science, doi: 10.1126/science.aav189). We followed the suggestion of the reviewer and deleted the HNF-1 β motif within KIRC_10767. We did not detect a PAX8 consensus motif within the PAX8 ChIP-seq signal. However, we used two sgRNAs to target the DNA-sequence located in the centre of the PAX-8 ChIP-seq peak. Both manoeuvres effectively introduced mutations within the binding-sites and led to reduced HIF-2 α mRNA which confirmed the functionality of these elements on HIF-2 α mRNA expression (Figure 5f-h, Supplementary Fig. 19, 20). We also explored whether HIF-binding to KIRC_10770 is dependent on the presence of the lineage-specific factors PAX8 and HNF-1 β . For this, we conducted HIF-1 β CUT&Tag-seq experiments in 786-0 cells depleted for PAX8 or HNF-1 β or depleted for the binding sites of both factors at *EPAS1*-enhancer E2. Under all conditions, we detected reasonably preserved HIF-1 β signals at the KIRC_10770 element irrespective of the presence of either of the two TF within the cell or at the enhancer. This indicates that HIF-binding to KIRC_10770 is not entirely dependent on PAX8 or HNF-1 β (Supplementary Fig. 21).

5. 5b lacks the impact of individual E1 and E2.

The reviewer is right. This experiment was conducted in primary ccRCC tumor cells which grew slow if growing at all and from which only a limited number of cells was available for experiments. This is why we transfected both sgRNA combinations together into these cells. The results were in line with results derived from combining the sgRNAs in 786-0 and RCC4 cells, in which we were able to also test individual sgRNAs. In the new version of the manuscript, we confirmed that both enhancers interact with HIF by HIF-1 β CUT&Tag-seq indicating that they were functional in these primary cells (Supplementary Fig. 13).

6. Is the SNP rs10193766 a risk site of ccRCC?

GWAS analyses have discovered a large number of RCC risk sites. However, rs10193766 or SNPs in high LD with this SNP have not been described as a risk site for RCC development. It was discovered to have an eQTL for HIF-2 α expression in the KIRC (ccRCC) data by PanCanQTL TCGA analyses (Gong, J. et al. 2018, Nucleic Acids Res, doi: 10.1093/nar/gkx861). However, in response to a valid comment of referee 2, we re-arranged the manuscript and removed the genotype-expression correlation for rs10193766.

In response to your comment and a comment from referee 2, we further evaluated the role of genetic susceptibility to renal cancer in regulation of *EPAS1* expression. To do this, we first accessed the TCGA-KIRC genotype and RNA expression data to test intronic variants which are associated with the development of renal cancer for effects on HIF-2 α expression. For the SNP rs11894252 which was reported in the initial GWAS and which was available for analysis on the genotyping array used by TCGA (Purdue, M. et al. 2010, Nat Gen doi: 10.1038/ng.723), we did not detect an eQTL for HIF-2 α mRNA (Supplementary Fig. 17b), which is in line with a previous report (Lasker, R.S. et al. 2019, Eur J Hum Genet., doi.org/10.1038/s41431-019-0455-9). However, we also evaluated rs12617313, which had been detected as an independent signal in a follow-up fine-mapping study and for which the risk allele was strongly associated with the presence of *VHL*-mutations in renal tumors (Han, S.S. et al. 2012, Human Molecular Genetics, doi.org/10.1093/hmg/ddr551). This SNP is a significant eQTL for HIF-

2 α mRNA expression in the TCGA KIRC tumor tissues, but not in non-diseased kidney tissue (Fig. 4d). Moreover, using our large cohort of PTC which we genotyped for this SNP, we detected a similar association of the rs12617313 genotype with HIF-2 α expression (Fig. 4e). Importantly, this association was only detectable in cells in which HIF was stabilized by DMOG, but not in untreated cells. A recent study also confirmed that this intronic region physically interacts with the upstream *EPAS1*-enhancer E1 indicating the existence of a complex interplay of enhancers and intragenic regions of *EPAS1* in renal cancer (Schmid, V. et al., 2019, Sci Rep, doi: 10.1038/s41598-019-55098-7). Thus, we can now link HIF-2 α expression with effects of ccRCC-associated germline variants and HIF-activated enhancers. This is the first report that explains some of the associations and effects of the intronic genetic variation within *EPAS1* in VHL-defective ccRCC biology. We have included these novel findings in Fig. 4d, e and Supplementary Fig. 17. We thank you for this important comment that has stimulated this new part of the work.

Reviewer #4 (Remarks to the Author):

We thank you very much for evaluating our manuscript as a co-reviewer.